



# Soil moisture redistribution and its effect on inter-annual active layer temperature and thickness variations in a dry loess terrace in Adventdalen, Svalbard

C. Schuh[1], A. Frampton[1,2], and H.H. Christiansen[3]

1 Department of Physical Geography, Stockholm University, Stockholm, Sweden
2 Bolin Centre for Climate Change, Stockholm University, Stockholm, Sweden
3 Department of Arctic Geology, University Centre in Svalbard, Longyearbyen, Norway

*Correspondence to*: andrew.frampton@natgeo.su.se

**Abstract.** High resolution field data for the period 2000-2014 consisting of active layer and permafrost temperature, active layer soil moisture, and thaw depth progression from the UNISCALM research site in Adventdalen, Svalbard, is combined with a physically-based coupled cryotic and hydrogeological model to investigate active layer dynamics. The site is a loess-covered river terrace characterized by dry conditions with little to no summer infiltration and an unsaturated active layer. A range of soil moisture characteristic curves consistent with loess sediments are considered and their effects on ice and moisture redistribution, heat flux, energy storage through latent heat transfer, and active layer thickness is investigated and quantified based on hydro-climatic site conditions. Results show that soil moisture retention characteristics exhibit notable control of ice distribution and circulation within the active layer by cryosuction subject to seasonal variability and site-specific surface temperature variations. The retention characteristics also impact unfrozen water and ice content in the permafrost. Although these effects lead to differences in thaw progression rates, the resulting inter-annual variability in active layer thickness is not large. Field data analysis reveals that variations in summer degree days do not notably affect the active layer thaw depths; instead, a cumulative winter degree day index is found to more significantly control inter-annual active layer thickness variation at this site. A tendency of increasing winter temperatures is found to cause a general warming of the subsurface down to 10 m depth (0.05 to 0.26°C/yr, observed and modelled) including an increasing active layer thickness (0.8 cm/yr, observed and 0.3 to 0.8 cm/yr, modelled) during the 14-year study period.

Keywords: Active layer; Permafrost; Numerical modelling; Circumpolar Active Layer Monitoring (CALM).

## 1 Introduction

Permafrost environments have been identified as key components of the global climate system given their influence on energy exchanges, hydrological processes, carbon budgets and natural hazards (Riseborough et al., 2008; Schuur et al., 2015). Over the last 30 years, air temperature in Polar Regions have increased by 0.6°C per decade, which is twice the global



average (IPCC, 2013). On Svalbard, long-term records indicate an increase in mean annual air temperature of 0.2°C per decade since the beginning of the 20th century (Humlum et al., 2011), and permafrost warming has been detected to a depth of 60 m based on borehole measurements (Isaksen et al., 2007). The relationship between climate and permafrost temperatures is, however, highly complex. Changes in active layer thickness can be buffered from effects of changing air

temperatures by vegetation, snow cover, and permafrost ice content and its thermal state, as well as by variable heat and water flows occurring in the active layer. The active layer is an important part of cold regions, where thermal and hydrological processes determine local phenomena such as erosion, hydrological and ecosystem changes (Karlsson et al., 2012; Lyon et al., 2009; Walvoord and Kurylyk, 2016; Walvoord and Striegl, 2007), and has implications for solute and carbon transport (Frampton and Destouni, 2015; Giesler et al., 2014; Jantze et al., 2013) and the global carbon-climate

feedback (Tarnocai et al., 2009). Also, the permafrost beneath the active layer limits percolation and subsurface water flow, allowing wet soils and surface ponding even in dry climates.

The thickness of the active layer is identified as an Essential Climate Variable and used as an indicator of permafrost degradation (GCOS, 2015), and the response of the active layer to climatic variations has been studied using climatic and ground thermal monitoring in several Arctic regions. For example, long-term air and ground temperature time series have

been evaluated as explanations for recent permafrost warming along a north-south transect through Alaska (Osterkamp, 2007). Lafrenière et al. (2013) found that the timing of snowmelt was a more significant factor controlling active layer temperature than snow accumulation at an arctic site in Canada, and Frauenfeld (2004) used ground temperature data collected at 242 stations across Russia to estimate thaw depths and identify long-term changes in active layer thickness. On Svalbard, Roth and Boike (2001) quantified the soil thermal properties and conductive heat fluxes for an experimental site

near Ny-Ålesund based on subsurface temperature data and soil moisture measurements. Akerman (2005) monitored active layer depths over several decades in the Kapp Linné area with regard to periglacial slope processes, and found active layer deepening to correlate well with increases in air temperature. Isaksen et al. (2007) reported rising permafrost temperatures with accompanying increases in active layer depths when evaluating thermal monitoring data for the 100 m deep Janssonhaugen borehole penetrating bedrock in central Adventdalen over a period of six years. For the period 2000-2007,

Christiansen and Humlum (2008) used a combined consideration of thermal monitoring data and Circumpolar Active Layer Monitoring (CALM) measurements to derive key controlling factors of active layer development at the UNISCALM study site in Adventdalen, central Svalbard.

The physics of thermal conduction and latent heat transfer has been the basis of several studies on permafrost dynamics and active layer processes (Hinzman et al., 1998; Kane et al., 1991; Shiklomanov and Nelson, 1999; Smith and Riseborough,

2010; Zhang et al., 2008). Studies have also proceeded beyond analysis of monitoring data to explore active layer dynamics; Westermann et al. (2010) used ground penetrating radar to identify soil moisture content and thaw depths at Ny-Ålesund, and Watanabe et al. (2012) applied electrical resistivity tomography to identify the seasonal variation in thaw depth in an alluvial fan in Adventdalen. Also, increasing air and permafrost temperatures in a 100 m deep borehole located in Tarfala,



Sweden have been observed and mechanistically linked through numerical modelling (Jonsell et al., 2013; Pannetier and Frampton, 2016), and a numerical study based on generic wet permafrost environments related to a site at Barrow, Alaska, indicates organic layer thickness and snow cover as key features controlling active layer thickness (Atchley et al., 2016). Several recent advances in permafrost and active layer model development have been made, in particular in the field of

coupled thermal-hydrogeological models of partially frozen ground (Bense et al., 2009; Karra et al., 2014; McKenzie et al., 2007; Painter, 2011). This has enabled studies on effects of permafrost degradation on changes in groundwater flows (Kurylyk et al., 2016; Scheidegger and Bense, 2014; Sjöberg et al., 2016), in particular showing expected increase in base flow and decrease in seasonal variability in discharge under warming (Frampton et al., 2013, 2011; Walvoord et al., 2012) and increased pathway lengths and delays in solute mass transport and breakthrough due to non-linear active layer thickness

increase (Frampton and Destouni, 2015).

In this study, a comprehensive long-term monitoring data set collected between 2000 and 2014 at the UNISCALM research site in Adventdalen, Svalbard, which includes ground temperature, soil moisture, and active layer thaw depth progression, is applied to a physically-based numerical model for partially frozen ground to investigate subsurface processes controlling active layer dynamics. This site is characterized by little precipitation and dry unsaturated conditions in the active layer. Soil

water retention is a critical but also highly uncertain parameter, which we investigate based on a range of characteristic retention curves commonly ascribed to the dominant sediment type (silt loam) of the location. The aim is to study soil moisture content and redistribution within the active layer and its effects on subsurface ground temperatures and inter-annual active layer thickness variations subject to site-measured ground surface temperature variations. Specifically, we study 1) How retention properties affect soil moisture and ice distribution and redistribution in the partially saturated active layer

under multiple freeze-thaw cycles consistent with site hydro-meteorological conditions; and 2) What the effect of different soil water retention properties on subsurface temperature and active layer thickness is over the course of the study period.

## 2 Site description

Adventdalen, located in the central part of Svalbard, is a typical U-shaped valley that dissects a landscape with peaks, ridges and plateaus. The valley is partially filled by periglacial sediments primarily in the form of colluvial, alluvial, and aeolian

deposits. During about four months of summer, mainly between June and September, the braided river system of Adventelva discharges into Adventfjorden (Killingtveit et al., 2003). The tributary streams draining to Adventelva have built up large alluvial fans on both sides of the valley, and several river terraces have been described that confine the braided channel system of Adventelva, and can extend up to several meters above river elevation (Bryant, 1982).

The UNISCALM site (78°12' N, 15°45' E) is located on a terrace on the southern side of Adventelva at an elevation of 10 m

a.s.l. (Fig. 1). The upper 1.3 m of sediment has been described as horizontally layered loess, i.e. silt-dominated aeolian sediment (Christiansen and Humlum, 2008). Information from adjacent boreholes and several study sites in the downstream



part of the valley supports this classification, and provides evidence of fine-grained, silt-dominated largely deltaic sediment with interbedded clay and sand down to a depth of 60 m (Gilbert, 2014). The site is sparsely covered by typical arctic tundra vegetation consisting of mosses and low vascular plants like *Salix herbacea* and sedges (Bryant, 1982).

Cryostratigraphic information obtained from drilling shows ice saturation in the top permafrost and considerable excess ice (up to 50%) at around 2.0-4.5 m depth in several boreholes in lower Adventdalen, with total carbon content estimated and measured to be only about 1-5% (Cable et al., In review), so that no considerable thermal insulation effects through soil carbon are to be expected (Koven et al., 2009). Despite its location at the outlet of the valley Endalen, the UNISCALM site is not notably influenced by surface runoff or lateral subsurface flows since it is cut-off from the upper part of the alluvial fan filling lower Endalen by a road construction. Endalselva drains the valley and discharges directly into the local water reservoir Isdammen, leaving the UNISCALM site unaffected from increased snow and glacial melt water runoff during spring and summer. Only snow which covers the site in end of winter and which does not sublimate may infiltrate the ground during snow melt.

Permafrost is continuous in Svalbard and can extend to 500 m depth in the mountains, whereas in the valley bottoms, such as in Adventdalen, the permafrost thickness is estimated to be about 100 m (Humlum, 2005). Ground temperature records from a 10 m deep borehole (ASB-2) adjacent to the UNISCALM site show that between 2009 and 2013, the mean annual ground surface temperature (MAGST) ranged from -13.1°C in March to +9.8°C in July (Juliussen et al., 2010); also the depth of zero annual amplitude was observed at -9.85 m depth with a mean temperature of -5.5°C, and with an average annual increase of 0.05°C/yr.

Svalbard airport is the nearest meteorological station located about 8 km northwest from the UNISCALM site (Fig. 1) and is closer to the sea and at a slightly higher elevation (28 m a.s.l.). Based on records from Svalbard airport (data from Norwegian Meteorological Institute, 2015) annual mean air temperature during the study period was -3.6 °C and mean annual precipitation was 195 mm; comparison with the latest climate normal 1981-2010 shows that the mean annual air temperature was noticeably lower (-5.1 °C) than during the study period (Tab. 1). Precipitation occurred mainly during winter and April, May and June were the driest months. Over the course of the study period, both precipitation and mean air temperature were subject to considerable inter-annual variations (Fig. S1).

## 3 Method

### 3.1 Field data

In this study, an ensemble of active layer thaw depth, active layer and permafrost temperature, and active layer soil moisture data collected at the UNISCALM site was used (Tab. 2). The UNISCALM site consists of a 100 x 100 m grid with 10 m spacing for thaw depth observations (Christiansen and Humlum, 2008). Measurements of thaw depth progression and active layer thickness were performed by probing all 121 grid points with a metal rod 8 to 15 times during the thaw season from



May to September each year (except for the year 2000, where only four measurements exist). Active layer temperature was measured by Tinytag miniature data loggers with temperature probes inserted directly into the sediment in a profile in the centre of the UNISCALM grid. The dataset encompasses hourly temperature measurements at the ground surface (0 m) and at 0.1, 0.2, 0.5, and 1.1 m depth. For the ground surface temperature measurements, two periods of missing data (31 August to 26 September 2001 and 26 April to 17 October 2004) were bridged using information from the next closest sensor at 0.1 m depth adopting a statistical data correction approach (Terink et al., 2010).

Near-surface permafrost temperatures and volumetric soil moisture content were recorded next to the UNISCALM grid. Hourly permafrost temperatures were available at 2.0, 3.0, 5.0, 7.0 and 9.85 m depth in borehole ASB-2 for the period between September 2008 and August 2014 (Juliussen et al., 2010). Volumetric soil moisture content was recorded using a PR2 profile probe (Delta-T) with sensing elements at 0.1, 0.2, 0.3, 0.4, 0.6, and 1.0 m depth. Soil moisture was registered with a 3-hour resolution for the period from July 2010 to August 2014. Temperature and soil moisture time series were converted into daily averages. Active layer thickness was considered both for the grid centre, which is the point nearest the location of the ground temperature measurements, as well as for the average of all grid points.

To quantify and evaluate the active layer response to thermal forcing, we consider a summer degree day index $\mathrm{SDD} = \sqrt{\sum_i T_i}$ defined as the square root of the sum of positive daily ground surface (i.e. at 0 m) temperatures $T_i$ from the onset of continuous thaw in spring until the start of active layer freeze-back in fall (thereby restricting summed days $i$ to a season essentially corresponding to summer). Complementary to SDD, we define the sum of daily ground surface temperatures between the summers $T_j$ as winter-degree days at the ground surface $\mathrm{WDD} = \sqrt{\sum_j T_j}$ and use it as an indicator for the thermal conditions preceding the respective summer thaw. The consideration of all summer degree days in this way has been proven useful to assess active layer response in particular to inter-annual temperature variability (Smith et al., 2009).

### 3.2 Numerical model

Simulations were performed with a recently developed numerical model, the Advanced Terrestrial Simulator (ATS), which can couple several thermal, hydrological and hydrogeological processes for heat flux and water flow applicable to partially frozen ground in cold regions (Atchley et al., 2015; Coon et al., 2016). The focus of our study is on effects of soil moisture retention characteristics on subsurface heat and moisture propagation in the active layer and permafrost for a dry and relatively flat site; hence the physics presently considered are hydrogeological heat and flow processes. This includes accounting for partitioning of water between ice, liquid, and vapour phases, phase-dependent thermal conduction, latent heat transfer, moisture migration due to phase change (wetting and cryosuction), and heat advection through the movement of water. The model solves a numerically discretized version of conservation equations for heat and water mass transport in porous media, adopting the unsaturated version of Darcy's law and accounting for phase partitioning by combined use of classical soil moisture retention curves and thermodynamic constraints derived from the Clausius-Clapeyron relation. Details of the underlying approach are provided in Frampton et al. (2011); Karra et al. (2014); Painter (2011). Since moisture



migration in the active layer is dominated by unsaturated flow for the site considered, and since retention characteristics are the main focus of this study, a brief summary of governing constitutive equations for phase partitioning as used by the model are provided in the following.

Partitioning between ice, liquid and vapour phase saturation, denoted by $s_i$, $s_l$, and $s_g$ respectively and constrained by $s_i +$

$s_l + s_g = 1$, is achieved by simultaneously inverting two constitutive relationships each relating liquid saturation with ice saturation as (Karra et al., 2014)

$$s_l = (1 - s_i) S_* (P_{cgl}) \tag{1a}$$

$$s_l = S_* [-\beta \rho_i h_0 \vartheta H(-\vartheta) + S_*^{-1}(s_l + s_i)] \tag{1b}$$

where $S_*$ is the retention curve for unfrozen liquid-gas phases, $P_{cgl}$ [Pa] is the liquid-gas capillary pressure, $\beta$ [-] is the ratio

of ice-liquid to liquid-air surface tensions, $\rho_i$ [kg m$^{-3}$] is the mass density of ice, $h_0 = 334$ [kJ kg$^{-1}$] is the enthalpy of fusion, and T [K] is temperature with $\vartheta = (T - T_0)/T_0$ [-] and $T_0 = 273.15$ K. The Heaviside function H is used to make Eq. 1b applicable to both frozen and unfrozen conditions. The first relation Eq. (1a) expresses a retention curve for unfrozen water where the available pore space is reduced by the fraction of ice present. The second relation Eq. (1b) describes liquid saturation as a function of total water content, where the first term in the square brackets corresponds to the capillary

pressure between ice-liquid phases when gas is absent (saturated conditions), and the second term is the addition to the ice-liquid capillary pressure when gas is present (unsaturated conditions). In our study, the retention curve $S_*$ is expressed using the van Genuchten (1980) model,

$$S_*(P_c) = S_r + (1 - S_r)[1 + (\alpha P_c)^n]^{-m} \qquad \text{if } P_c > 0 \tag{2a}$$

$$S_*(P_c) = 1 \qquad \text{if } P_c \le 0 \tag{2b}$$

combined with the Mualem (1976) model for rescaling liquid phase permeability,

$$k_{rl} = (s_l)^{\frac{1}{2}} \left[ 1 - \left( 1 - (s_l)^{\frac{1}{m}} \right)^m \right]^2 \tag{3}$$

where $S_r$ [-] is residual saturation, $P_c$ [Pa] capillary pressure, and $\alpha$ [Pa$^{-1}$] and $m = 1 - 1/n$ [-] are model parameters. The exponent m controls the shape of the soil moisture retention curve and can be related to the pore size distribution of the texture (Mualem, 1976; van Genuchten, 1980), where larger values generally correspond to smaller pore size variability, i.e.

to well-sorted textures.

### 3.3 Model configurations

The model domain was configured as a 1D vertical column of the subsurface, with top corresponding to the ground surface and bottom at a depth of 10 m, and with a 0.1 m resolution of cell heights. A depth of 10 m was chosen since this corresponds to the approximate depth of zero annual amplitude measured at the UNISCALM site. Given the comparatively

homogeneous soil composition at the site, the entire model domain was assigned the physical and thermal properties of silt loam (Tab. 3). Then different soil moisture retention characteristics were investigated by varying the parameters $\alpha$ and $m$ of



the van Genuchten model Eq. (2), all within bounds applicable to silt loam (Tab. 4 and Fig. 2). In all scenarios, residual saturation $S_r$ is set to zero, consistent with commonly adopted practice for silty soils (Destouni, 1991; Painter and Karra, 2014; Wang et al., 2015; Watanabe and Wake, 2009; Weismüller et al., 2011). A reference case is first defined using average values ($\alpha=8\cdot10^{-4}$ Pa$^{-1}$ and $m=0.19$) obtained from the UNSODA soil hydraulic database (Ghanbarian-Alavijrh et al., 2010) for silt loam soils. Then, both $\alpha$ and $m$ were varied independently by ± 50% with respect to the reference case, resulting in model scenarios labelled as 'high alpha' ($\alpha=12\cdot10^{-4}$ Pa$^{-1}$; average $m$), 'low alpha' ($\alpha=4\cdot10^{-4}$ Pa$^{-1}$; average $m$), 'high m' ($m=0.29$; average $\alpha$), and 'low m' ($m=0.1$; average $\alpha$). Two additional scenarios, 'max case' and 'min case', represent the high and low ends of the parameter range by combining high alpha with high m, and low alpha with low m, respectively. All of these parameter values are within the range of the variation for silt loam reported in the UNSODA database and in literature (Destouni, 1991; Wang et al., 2015; Watanabe and Wake, 2009).

To enable systematic investigation of the different soil retention properties with initial conditions consistent with site conditions, each simulation case was initialized with unsaturated frozen ground to a depth of about -1.2 m and saturated below. This yields a system with ice-liquid-vapour in the upper part and ice-liquid in the lower, where the phase partitioning differs for the respective scenarios and is controlled by their respective retention curves. Thereafter an annually periodic steady state was obtained by running multiple iterations (i.e. a spin-up) using a smoothed version of the observed ground surface temperature time series as surface boundary condition. The convergence criteria was set as a maximum temperature difference between two consecutive iterations to be less than 0.1°C. This yielded an active layer thickness, defined by the 0˚C-isotherm, of approximately 1 m for most cases, which is also consistent with site conditions.

The simulation investigations were then carried out by directly adopting the daily ground surface temperature time series (i.e. temperatures measured at 0 m over the time period 2000-2014) as the top boundary condition of the model. The bottom boundary was assigned a linear increasing temperature trend from -6.05 °C (September 2000) to -5.4 °C (August 2014), consistent with the increasing temperature trend of 0.05 °C/yr observed for the time period 2008 until 2014 at depth -9.85 m. Based on the local topographical setting with no drainage to the study site, field evidence for a shallow snow cover of 20-30 cm (Christiansen and Humlum, 2008), high wind erosion from unstable snow layers and high evaporation and sublimation rates (Westermann et al., 2010), combined with previous modelling studies indicating that inter-annual active layer variation at the site is largely unaffected by infiltration derived from the small amounts of spring snowmelt and summer precipitation (Schuh, 2015), infiltration was determined to be of minor importance and hence not further considered in the present study.

The simulated results for the different cases were compared against field measurements considering active layer thickness, subsurface ground temperature and soil moisture. The simulated active layer thickness was defined as the deepest numerically calculated 0˚C-isotherm for each year. Differences between measured and simulated ground temperatures were quantified by a root mean squared error RMSE, averaged over a time period corresponding to the length of the available time series as



$$RMSE = \sqrt{\frac{1}{N}\sum_{i=1}^{N}(S_i - O_i)^2} \qquad (4)$$

where $S_i$ and $O_i$ are the simulated and observed daily temperatures, respectively, and $N$ is the number of data points.

## 4 Results

### 4.1 Active layer thickness and subsurface temperatures

Model results show that the different assumptions regarding soil water retention properties impact subsurface temperatures and active layer thickness (ALT). The modelled ALT ranged from 90 to 130 cm considering all scenarios and years (Tab. 5). Cases with a small value for the parameter $m$ ('low m' and 'min case') generally had ALTs in the range 100-130 cm, somewhat larger than moderate or high values of $m$, which had ALTs in the range 90 to 110 cm (e.g. 'ref case' and 'high m'). Accordingly, the mean ALT over the study period was 118-119 cm for the cases assuming a small value for $m$ and 99-103 cm for the other cases. Overall, the smallest ALT were simulated for the years 2005 and 2008, and the largest for the years 2006, 2013, and 2014. This can be compared to the measured thaw depth where the UNISCALM grid average ranged from 74 to 110 cm depending on year, and the smallest occurred in 2005 and the deepest in 2008, and with a mean of 98 cm over the entire time period (Tab. 5).

Inter-annual variation in simulated ALT was comparatively small for most of the scenarios with standard deviations of about 5-6 cm. For scenarios with low value for $m$, inter-annual variation in ALT was slightly higher with standard deviations of about 8-9 cm. The standard deviations based on measured thaw depth were close to 8 cm, both for the grid average as well as only considering the centre probing location. Depending on scenario, a trend of increasing ALT varied from 0.3 to 0.8 cm/yr for the entire study period, which can be compared to the trend of 0.8 cm/yr obtained from the UNISCALM grid measurements average (Tab. 5).

Active layer temperatures recorded at the UNISCALM site were generally mimicked best by scenario 'max case' with a depth-averaged root mean squared error (RMSE, Eq. 4), of 0.07°C in the active layer (Tab. 6). Most other scenarios represented summer temperatures well but generally underestimated winter temperatures (not shown), resulting in mean RMSE values up to 0.18°C (Tab. 6, e.g. case 'low alpha'). Note also RMSE values are generally smaller at the intermediate depths (0.2 m and 0.5 m) and larger near the surface (0.1 m) or near the top of permafrost (1.1 m). In the permafrost, the simulations generally overestimated the seasonal temperature amplitude both in summer and winter, but to a lower degree than in the active layer (not shown). In the permafrost, the depth-averaged RMSE ranged between 0.05°C in scenario 'high alpha' to 0.10°C in scenario 'low alpha' (Tab. 6).

The observed tendency of increasing temperatures over the study period was represented well by the simulations, irrespective of the chosen soil water retention parameters, where the average of all scenarios resulted in a trend of 0.25°C/yr at 0.1 m depth, and decreasing to 0.05°C/yr at 9.85 m depth, consistent with the corresponding trends obtained from the



measured active layer and permafrost temperatures. Also, note that the observed increase in ground surface temperature can be ascribed exclusively to increasing winter temperatures, since the trend in summer temperature on the ground surface was essentially stable (-0.05°C/yr) over the study period, in contrast to the relatively strong increase in winter ground surface temperature (+0.25°C/yr).

## 4.2 Ice and water content

By model design, the permafrost was fully saturated below a depth of about 1.2 m in all scenarios. However, depending on the retention curve parameterization the different scenarios exhibit different fractions of ice and unfrozen water content in the permafrost. The scenarios with large $m$=0.29 ('high m' and 'max case') have greatest volumetric ice contents of 39%, followed by scenarios with intermediate $m$=0.19 ('high alpha', 'ref case', and 'low alpha') with approx. 36% ice content, and scenarios with $m$=0.1 ('low alpha' and 'min case') with only 26% ice content (Fig. 3). The remaining fraction to full saturation, i.e. 1%, 4%, and 14%, respectively, consists of unfrozen liquid water (porosity was assigned to 40%, cf. Tab. 3, consistent with field conditions).

Although no additional infiltration is imposed so that the total water mass in each modelled systems is constant over time, the different retention curves cause different initial phase partitioning so that the various model cases may differ in total water mass. During freezing of the active layer, water becomes repartitioned according to phase state and migrates by cryosuction, and both ice content and distribution vary considerably between scenarios (Fig. 3). Total ice content is observed to be mainly a function of α in the retention curve (Eq. 2) with lower values resulting in higher ice content; the largest total ice content was obtained for the case 'low alpha' (30-35%), followed by 'ref case' and 'min case' (25-30%), 'high m', 'low m', 'high alpha' (15-25%), and 'max case' (12-18%). However, the ice distribution and spatial layering with depth was mainly a function of the value of $m$. For low values $m$=0.1 no particular ice layering was observed, instead the ice distribution was relatively homogeneous throughout the active layer (Fig. 3c). For the other cases, layers of increased ice content developed more clearly, typically with a tendency of a narrow layer of higher ice content just below the ground surface as well as within a layer above the permafrost table. The layering was most pronounced where $m$ was highest ($m$=0.29; Fig. 3d). Note also the vertical position and depth extent of the layers vary over time for all cases, indicating different ice redistribution occurring each freeze-up season.

During summer, the simulated water content in the approximate centre of the active layer, i.e. at 0.6 m depth, was between 15-38% depending on model scenario (Fig. 4c), with slight increase with depth (Fig. 4d). The increase in water content generally coincided with the onset of thaw in spring and with the location of the thawing front. Note the timing of increase and subsequent drop of the water content deteriorates with depth. Except for scenarios assuming a small value $m$=0.1 ('low m' and 'min case'), the model cases show a delayed increase and a premature and prolonged decrease in water content in the lowermost active layer (i.e. at 1.0 m depth, Fig. 4d).



Moderate peaks in measured water content can be seen in the field measurements at depth 0.1 m (Fig. 4b, black line), which are significantly attenuated already at 0.6 m and 1.0 m depth (Fig. 4c and d, black line). These could correspond to brief infiltration events, for example caused by snow melt at onset of thaw, confined to the uppermost soil layers, while parts below remain frozen (cf. temperatures at 0.5 m and 1.1 m depth, Fig. 4a), thereby momentarily saturating the top part of the

active layer. Aside from the peaks at 0.1 m depth, the scenarios 'max case' and 'high m' reflect the observed total water content in the thawed active layer (15-25%) to a high degree, whereas the other scenarios (e.g. 'min case') overestimated it considerably. In winter, the amount of unfrozen water seems to be best represented by scenarios using an average value $m=0.19$, i.e. 'ref case', 'low alpha', and 'high alpha'.

## 5 Discussion

### 5.1 Soil moisture and ground ice distribution

Even for the same soil type considered (silt loam), the specific choice of retention parameters consistent within the range applicable for that texture class, as exemplified by the seven simulation cases studied here (Tab. 3), clearly has an influence on the amount and redistribution of both liquid water and ice in the active layer with its annual freeze-thaw cycles (Fig. 3 and Fig. 4). The water retention parameters tested in the different scenarios also impacted the unfrozen water content in the

active layer as well as in the permafrost. The parameter $\alpha$ normalizes capillary pressure (Eq. 2) so that the main effect of a decrease or increase in $\alpha$ results in a general shift of the retention curve up or down, resulting in a higher or lower overall liquid saturation, respectively (cf. Fig. 2). This parameter is varied by almost one order of magnitude between the different cases, causing a maximum difference in water content of about 10% in the active layer (corresponding to a change of about 25% in terms of saturation).

The parameter $m$ controls the overall shape and slopes of the retention curve. This way, it is significant for cryosuction and thus redistribution of ice in the active layer during freeze up. The retention curves with small values $m=0.1$ exhibit relatively smooth overall slopes even for low saturations (cf. Fig. 2), and do not result in high cryosuction effects. Instead, scenarios with high $m=0.29$ and comparatively steeper slopes for low saturations result in the greatest changes in capillary pressure and hence greater cryosuction. A reason for this is the respective fraction of water that remains unfrozen even at

temperatures below 0°C. Given a certain soil water content in the active layer, a drop in subsurface temperature of about 10°C, as typically observed in winter, results in an increase in capillary pressure since liquid water undergoes a phase change to ice. The liquid water content is thereby reduced, resulting in an effective drying out of the pore space and yielding a hydraulic gradient towards the freezing front. For retention curves with high $m$ value displaying a higher non-linearity at low saturations, this increase in capillary pressure leads to a small fraction of unfrozen water (~1% for $m=0.29$). In contrast, for

the same increase in capillary pressure, retention curves with low $m$ value yield higher unfrozen water contents (~14% for





$m$=0.1). As a consequence, cryosuction and the resulting moisture migration to the freezing front is greater for scenarios assuming higher $m$ since such cases undergo greater phase change and greater effective drying of the pore space.

This is reflected in the simulated ground ice pattern redistribution, where scenarios with high $m$ values ('high m', 'max case') typically showed well-defined layers of ice in the upper and lower parts of the active layer, whereas cases with low $m$ values ('low m', 'min case') did not exhibit such a distinct variable ice pattern (Fig. 3c). The parameter $m$ relates to physical soil properties, specifically the pore size distribution and pore connectivity/tortuosity, where large $m$ is generally considered to correspond to a small variability in pore size distribution, i.e. corresponding more to a well-sorted texture. Note also the parameter $m$ controls relative permeability in the Mualem model (Eq. 3 and Fig. 2, inset) and hence hydraulic conductivity. At the UNISCALM site, the sediment has been described as a well-sorted loess sediment and thus may be more consistent with larger $m$ values. Furthermore, the comparison against soil water content in the active layer (Fig. 4) revealed that scenarios using an average $m$=0.19 or large $m$=0.29 seemed more consistent with observed unfrozen water content during the frozen period. In scenario 'max case' the soil water content during the thaw period is lowest of all scenarios, and at a level which is consistent with field measurements (~10-20%).

Based on simulated results, we infer that the higher ice content in the top-most parts of the active layer developed by cryosuction caused by the downward moving freezing front, whereas the ice in the deeper parts of the active layer may be a combination of cryosuction-induced moisture migration together with water percolating down by gravity during the thaw period. Here the cryosuction-effect may either counteract percolation or enhance it, depending on if one or two-sided freezing occurs. Considering the downwards-moving freezing front from the ground surface, this should cause a cryosuction-induced moisture migration towards the freezing front, i.e. move moisture upwards and thereby work against gravity-driven percolation flow. A freezing front moving upwards from the top of permafrost would, however, cause a cryosuction-induced moisture migration downwards, thereby working together with percolation flow. For two-sided freezing, the rates of heat propagation, which in turn would also depend on the variability of the surface temperature and the thermal state of the permafrost below, would govern the strength of cryosuction-induced moisture migration from the respective fronts and their net effect on moisture movement combined with percolation in the active layer. Considering the general spread of increased ice content at approximate depths 0.5-1.0 m (e.g. Fig 3a), cryosuction-induced moisture migration may be occurring in both directions, i.e. consistent with two-sided freezing, at least for cases with moderate to high values of $m$.

Field investigations indicate downward freezing from the ground surface may cause ice lenses to form mainly in the upper active layer, and two-sided freezing may cause segregated ice and a dryer middle active layer to form (French, 2007). Also, an augmented amount of ice lenses near the bottom of the active layer and top of permafrost is generally interpreted as ice segregation through upward freezing from the permafrost table or caused by percolation and moisture migration down to the permafrost (Cheng, 1983; Mackay, 1972). At the UNISCALM site, lenticular cryostructures at the top of the permafrost have been observed (Cable et al., In review; Gilbert, 2014), and recent sediment core retrieval in March 2015 on a river terrace near the UNISCALM site showed increased ice lenses in the upper active layer as well as at the top of the permafrost,





whereas the middle active layer was comparably dry showing further evidence for both downward and upward freezing. Boike et al. (1998) identified two-sided freezing at a comparable study site in Siberia using frost probing and water content information. It therefore seems likely that two-sided freezing occurs at the UNISCALM site, even if the downward freezing component seems to be dominant. Also the simulation cases considered here are consistent with these general observations

of two-sided freezing, at least for cases with moderate to high $m$ values, corresponding to smaller pore size variability and hence well-sorted textures.

## 5.2 Thaw progression and refreezing

Soil moisture and ground ice distribution impact subsurface temperatures and temperature gradients since heat propagation into and out of the ground is largely controlled by both phase-dependent thermal conductivity and latent heat transfer. The

10 model results showed that scenarios assuming a small $m$ value ('low m', 'min case') resulted in greater ALT and greater inter-annual variations in ALT as well as larger increasing trends of ALT over the study period (Tab. 5). The timing and duration of thaw in the upper active layer was essentially the same for all cases (Fig. 4b), whereas it was delayed and underestimated in the lower active layer for cases with large $m$ value (Fig. 4d). A more detailed consideration of the thawing process, here exemplified for year 2011 (Tab. 7), further showed that to thaw the upper 1 m of the model domain, 58-60 days

were required in scenarios using small $m$ values ('low m', 'min case'), 74-76 days in scenarios using average $m$ values ('ref case', 'high alpha', 'low alpha'), and 78-85 days in scenarios using large $m$ values ('high m', 'max case'). This is altogether evidence for a more efficient heat propagation into the ground for scenarios assuming a low $m$ value, and thereby a clearer reflection of the influence of ground surface temperature dynamics on the deeper subsurface.

Tab. 7 also helps understand the different thaw progression rates by summarizing the prevailing moisture conditions in the

20 upper 1 m of the model domain (essentially corresponding to the active layer) at the end of winter and just prior to the onset of thaw in 2011. An effective thermal conductivity $\kappa_e$ was calculated based on the fraction of ice, water, and air content (excluding the soil matrix) at that point in time, and the total latent heat $L$ was determined directly from the respective ice mass. Accordingly, effective thermal conductivities clearly reflect the phase partitioning, with lowest $\kappa_e$=0.34 and $\kappa_e$=0.47 J kg$^{-1}$ K$^{-1}$ found for settings with largest air fractions of 24% ('max case') and 18% ('high m') respectively, and highest

$\kappa_e$=0.73 J kg$^{-1}$ K$^{-1}$ occurring for a nearly ice-saturated setting with small air content of 3% ('low alpha'). The latent heat buffer depended on simulation case and ranged from $L$=20 ('max case') to $L$=42 MJ ('low alpha'). The thaw rate is however highest in scenarios 'low m' and 'min case', which are combining medium to high effective thermal conductivity ($\kappa_e$=0.54 to 0.60 J kg$^{-1}$ K$^{-1}$) with medium to low latent heat consumption ($L$=27 to 31 MJ). In contrast, as shown in scenario 'max case', heat propagation can be severely hindered by high air content resulting in low effective thermal conductivity, even if it only

contains a small amount of ice ($L$=20 MJ). For the active layer freeze-back in autumn the same processes apply in reverse, i.e. heat flux is inverted flowing from the subsurface up and out of the ground. For scenario 'max case', this implies a slow progression of the freezing front, visible in the gentle drop of liquid water content simulated in the lower active layer (Fig.





4d). However, upward freezing from the permafrost table causes an earlier freeze-back of the active layer bottom, so that, despite a less efficient heat transfer in some cases, active layer freeze-back is completed at about the same time in all scenarios.

## 5.3 Inter-annual active layer variation and permafrost development

Despite the differences in heat propagation through the subsurface and the resulting greater ALT in scenarios using high $m$ values, the inter-annual variation in ALT over the study period follows a consistent pattern irrespective of soil water retention properties. The large inter-annual air temperature variations in Svalbard are mainly due to the extreme maritime location with particularly high temperature fluctuations during the long Arctic winter (Humlum et al., 2003). Accordingly, the winter WDD index (cf. Sect. 3.1) shows a high variability over the study period and ranges between -40 and -54 (with coefficient of variation CV=0.09), compared to relatively stable summer conditions where SDD ranges between 26 and 31 only (CV=0.04) (Fig. 5a-b and Fig. 6), consistent with similar calculations reported by Christiansen et al. (2013). The field measurements of thaw depth show comparatively small inter-annual variations, except for the years 2005 and 2008 where probing of the CALM grid identified exceptionally small and large thaw depths of 74 cm and 110 cm, respectively (Fig. 5a-b and Tab. 5, grid average). Recall also the ground surface temperature measurements indicate essentially constant or slightly cooling summer temperatures with trend -0.05 ˚C/yr but a notable winter warming trend of +0.25˚C/yr over the time period 2000-2014.

Note that the smallest thaw depth during the study period, measured in 2005, occurred after three consecutive and comparatively cold winters 2002-2004 (Fig 6). The greatest thaw depth, measured in 2008, occurred after two moderate and warm winters 2006 and 2007. Although the active layer (by definition) responds to the current year's summer warming, such decreased versus increased temperature changes from previous years will affect the subsurface thermal state by depth lags in heat propagation and energy storage. Note also that although the SDD index in 2005 is rather low, consistent with the shallow active layer occurring that year, it is also low in 2008, which is inconsistent with the deep active layer occurring in 2008. This, combined with the previously mentioned observation that the SDDs vary much less than the WDDs, supports the proposition that WDDs and hence winter temperature and duration are the more dominant factors controlling active layer thickness at the UNISCALM site on Svalbard.

A mechanistic explanation for this is that active layer thickness will in general not respond symmetrically to colder versus warmer surface conditions. Colder summers can directly cause a decrease in ALT by not providing sufficient heat for a typical thaw depth, as seen in 2005, whereas an increase in ALT by warmer summers can be impeded by ice-rich conditions at the permafrost table causing a latent heat buffer. Thus, a relatively moderate increase in summer thermal conditions as observed by the SDD index may not necessarily provide sufficient heat to thaw fully or near-fully ice saturated permafrost. Such a buffering function of an ice-rich upper permafrost conforms to the concept of a transition zone (Shur et al., 2005). Accordingly, latent heat buffer effects may counterbalance thaw in warmer years, so that the transition zone increases overall



thermal stability in the underlying permafrost. Only after a sufficient number of consecutive warm winters or years would such a transition zone eventually degrade enabling the active layer to deepen.

In contrast to the field observations, the simulated ALT showed a more pronounced correlation to SDD (with $R^2$=0.41 for 'max case') and a comparatively weak correlation to WDD (with $R^2$=0.19 for 'max case') (Fig. 5c-d). The predominant effect of summer conditions is reflected in the shallow ALT that have been simulated for the years 2005 and 2008 (Tab. 5). Both these years were characterized by cold summers with SDD = 26 and 27, and preceded by average winters with WDD = -47 and -45, respectively. The large thaw depth in 2008 is however not captured by the models. Assuming that the thaw depth occurring in 2008 is caused by heat storage in the permafrost, this could indicate that thermal storage effects are not correctly captured by the current setups.

Several studies have identified significant correlation between summer degree days and active layer depths (e.g. Christiansen, 2004; Smith et al., 2009). However, Osterkamp (2007) found warming winters to be a main cause for increasing permafrost temperatures in the Arctic Coastal Plain, Alaska, while summers even showed a slight cooling. The importance of winter conditions for thaw during the subsequent summer has also been emphasized by Burn and Zhang (2010). For their study site in the Mackenzie Delta, Canada, they found that observed variation in ALT could only partially be explained by the varying summer temperatures and they showed that ALT was also influenced by preceding winter conditions because of a change of energy components entering the ground. After warmer winters less energy was required to warm the subsurface (sensible heat), so that more energy could be used to thaw the ground (latent heat). Wintertime snow cover thickness and duration has been shown to exhibit control over the ground thermal regime (Lafrenière et al., 2013). Also, Mackay and Burn (2002) investigated 20 years of active layer development after the artificial drainage of Lake Illisarvik in arctic Canada, where they identified the warming of the subsurface following increases in snow depths as the major controlling factor for the observed variation in ALT as opposed to summer weather conditions.

The limited effectiveness of increased summer temperatures for increasing ALT identified here for the UNISCALM site calls for a more cautious consideration of thermal influencing factors. In the Arctic, the future temperature rise is expected to be most pronounced during winter and precipitation is expected to increase (IPCC, 2013) especially as snow during autumn and winter (Kattsov et al., 2005). Increased snow thickness and/or duration should affect the ground thermal regime such that progressive active layer deepening and permafrost warming may be expected even if summer temperatures remain stable or decrease. At the same time, the effect of active layer deepening may lag behind surficial warming by several years if an ice-rich transition zone near the permafrost table exists.

## 6 Summary

The active layer dynamics at a dry loess-covered site on Svalbard was studied based on a high-resolution field data set combining active layer and permafrost temperatures, soil moisture, and active layer thaw depth in conjunction with a



physically based coupled permafrost-hydrogeological model. Simulations were configured based on site conditions and then used to investigate how different soil moisture retention curves consistent with the silt loam sediment texture observed at the site impacted active layer thickness, heat propagation and subsurface temperatures, as well as moisture and ice redistribution by wetting and cryosuction. The main conclusions related to the stated investigation questions are:

- Even when constrained to a single sediment type (silt loam), the specific choice of retention parameter values leads to different moisture and ground ice distributions. In particular, well-sorted textures with small pore size variability (large exponent $m$ in the van Genuchten retention curve) lead to more distinct subsurface ice heterogeneity.

- The seasonal temperature variability during freeze-up exhorts significant control on cryosuction leading to inter-annual differences in subsurface ice heterogeneity

- The choice of retention parameter values also impacts thermal properties through phase partitioning, as well as impacting the amount of unfrozen water content at temperatures below freezing, resulting in different latent heat consumption and heat propagation rates in both the active layer and permafrost.

- Uncertainty in retention parameters can have significant impacts on predicted thermal development as well as ice distribution and water migration.

- The simulations showed that the highest thaw rates occur for textures which result in medium to high effective thermal conductivity and medium to low latent heat consumption, corresponding to poorly-sorted textures with large pore size variability (small exponent $m$). These high thaw rates also led to both greater active layer thickness, and slightly greater inter-annual variations in active layer thickness.

- Active layer thickness, as observed in the field, responded primarily to the cumulative temperature during the preceding winters, as opposed to cumulative summer temperatures during thaw.

- Deepening of the active layer at this site is mainly controlled by consecutive years of winter warming; an immediate response is impeded by a latent heat buffer caused by ice-rich conditions near and below the permafrost table.

## Acknowledgements

This study was funded by the Swedish Geological Survey SGU (project 362-1593/2013). Partial support from Stiftelsen Lars Hiertas Minne is also gratefully acknowledged. We would like to thank Ethan Coon at Los Alamos National Laboratory, NM, USA, for valuable assistance in using the ATS code, and Georgia Destouni at Stockholm University for providing scientific feedback. Furthermore, we thank numerous field assistants for their efforts in collecting measurements at the UNISCALM site.



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



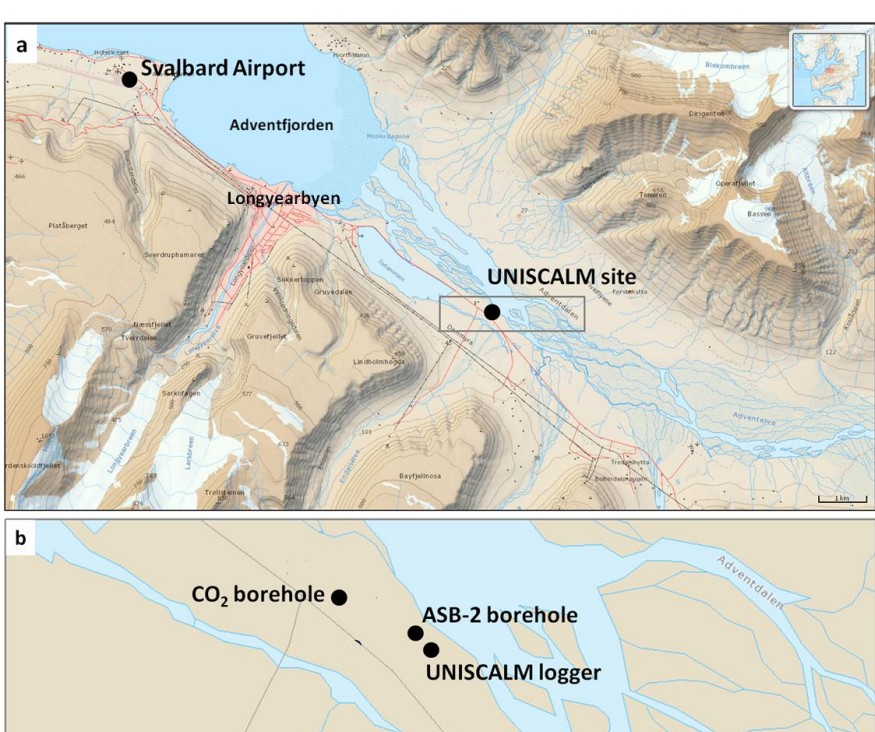

Figure 1: Lower Adventdalen in central Spitsbergen, Svalbard, showing the location of the UNISCALM site. The meteorological station is located at Svalbard Airport (Data: Norwegian Polar Institute).





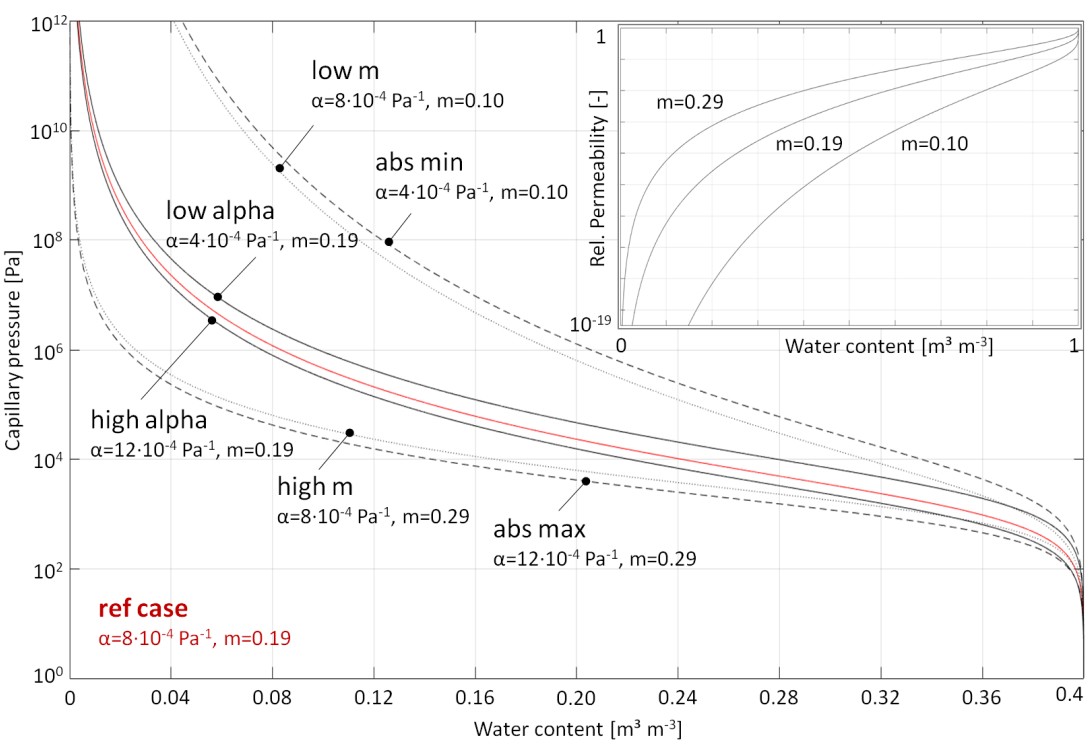

Figure 2: Retention curves $S_*$ described by the van Genuchten model (Eq. 2) and as used in the different model scenarios; residual saturation set to zero for all cases. (Inset) Relative permeability described by the Mualem model (Eq. 3).





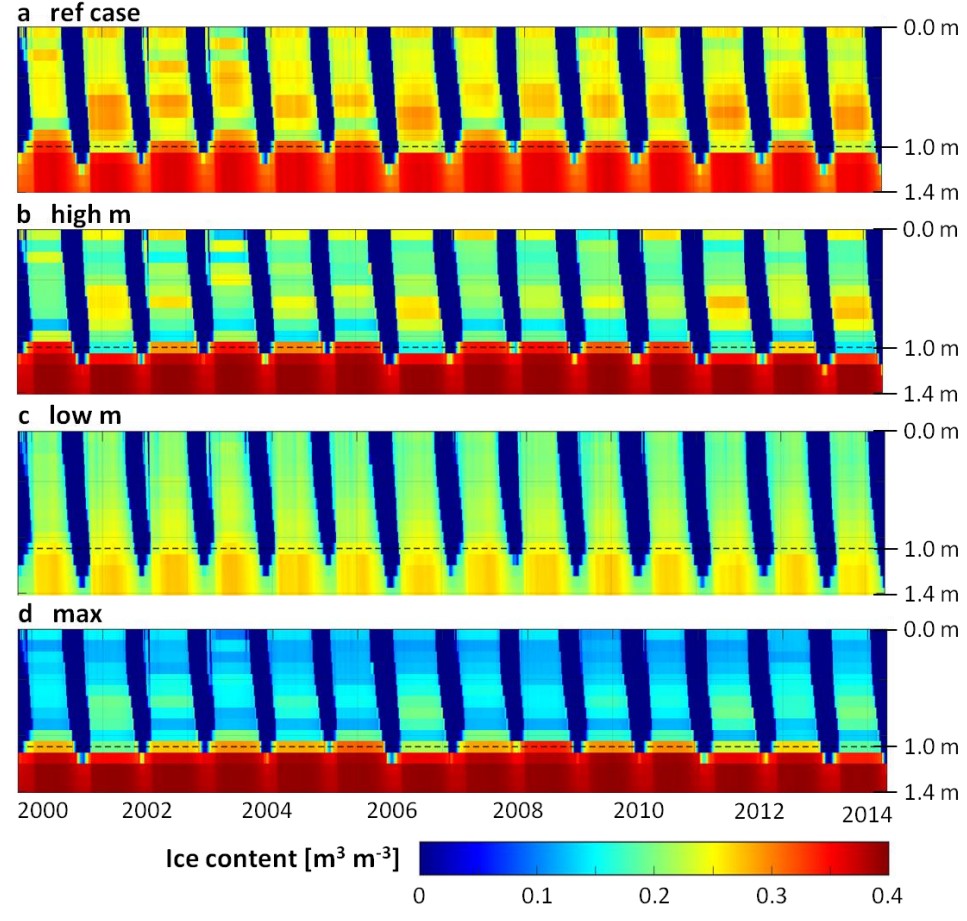

Figure 3: Ice content [m³ m⁻³] in the active layer and upper permafrost over the study period as simulated in four selected scenarios.



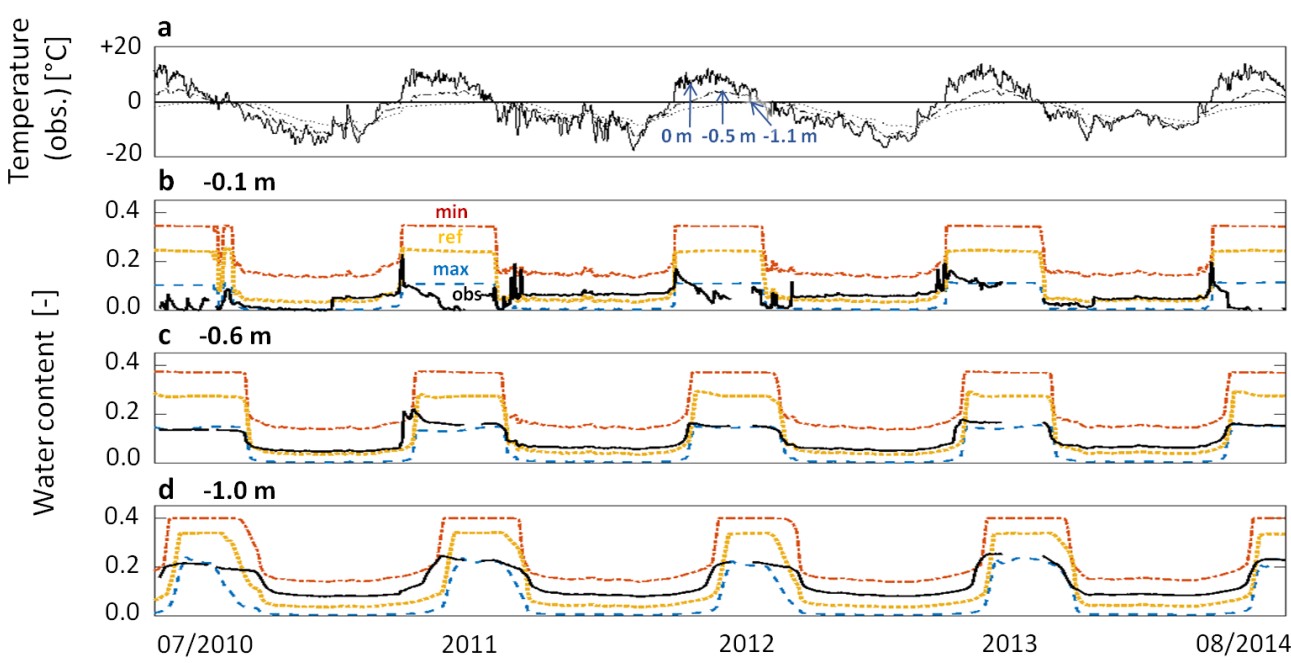

Figure 4: (a) Daily measured ground temperatures on the surface 0 m (solid) and at depths 0.5 m (dashed) and 1.1 m (dotted) between July 2010 and August 2014. Also volumetric water content at depths (b) 0.1 m, (c) 0.6 m and (d) 1.0 m as observed (black) and modelled (showing ref case, min case and max case in yellow, red and blue respectively).



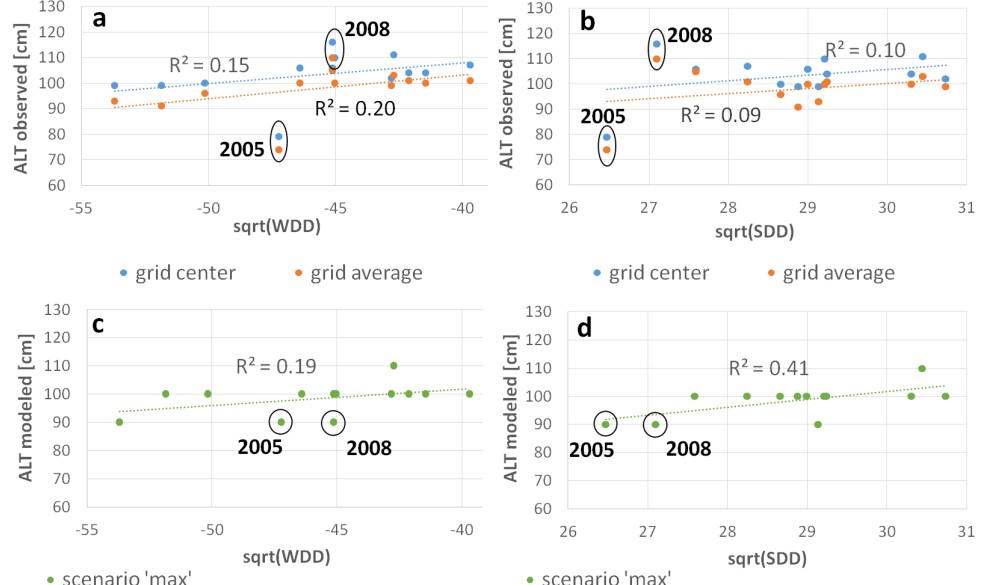

Figure 5: (a,b) Correlation of observed and (c,d) modelled active layer thicknesses to thermal conditions at the ground surface, considering winter degree days (WDD) and summer degree days (SDD) separately. Observed active layer measurements include information both from the grid centre (blue) and the grid average (orange). Modelled active layer thickness are shown for scenario 'max case' only (green).





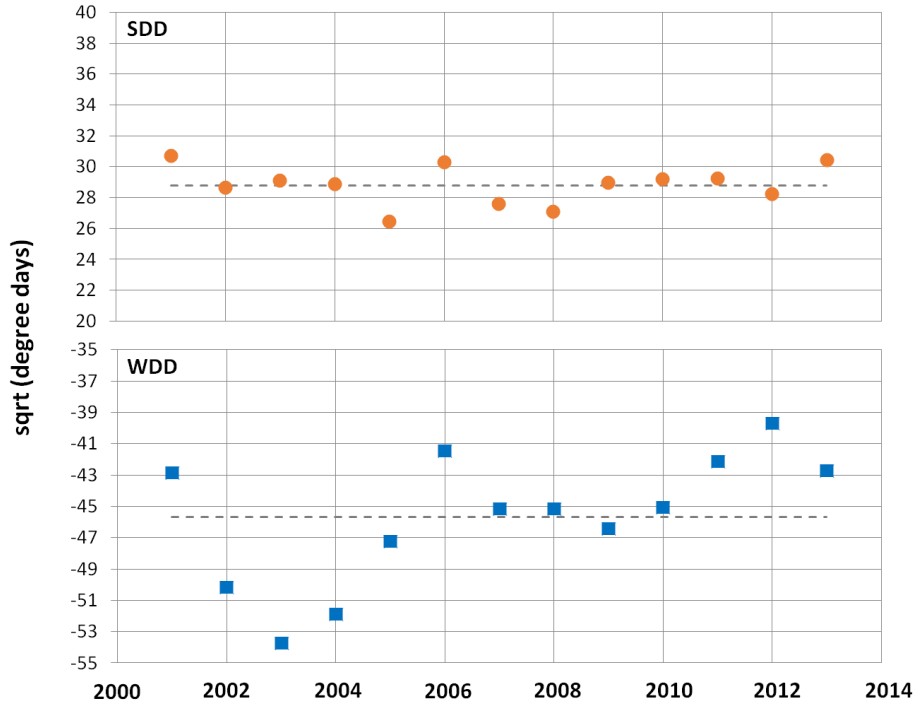

Figure 6: Summer-degree days SDD (circles) and winter-degree days WDD (squares) as an indicator for the thermal conditions at the ground surface over the course of the study period. The dashed lines indicate the respective means over the time period.



Table 1: Mean monthly and annual temperature [°C] and precipitation [mm] at Svalbard airport during the study period 2000-2014; statistics (mean Ø and annual sum Σ) for the latest climate normal 1981-2010 are included for comparison (Data: Norwegian Meteorological Institute).

|  | 2000-2014 |  |  |  |  |  |  |  |  |  |  |  |  |  | 1981-2010 |
|---|---|---|---|---|---|---|---|---|---|---|---|---|---|---|---|
|  | Jan | Feb | Mar | Apr | May | Jun | Jul | Aug | Sep | Oct | Nov | Dec |  |  |
| **T [°C]** | -9.5 | -10.0 | -12.9 | -8.7 | -1.8 | +3.7 | +7.0 | +6.2 | +2.1 | -3.5 | -6.7 | -8.5 | Ø -3.6 | Ø -5.1 |
| **P [mm]** | 20 | 12 | 15 | 9 | 8 | 7 | 21 | 20 | 22 | 21 | 19 | 22 | Σ 195 | Σ 187 |

Table 2: Field data used (Data: UNIS, NORPERM).

| Method | Data | Period |
|---|---|---|
| *Tinytag* individual thermistor probes connected to miniature temperature loggers | Ground surface temperature (0.0 m) [°C]<br>Subsurface temperature (-0.1, -0.2, -0.5, -1.1 m) [°C] | 01.09.2000-31.08.2014 |
| *GeoPrecision* thermistor string with data logger | Subsurface temperature (-2.0, -3.0, -5.0, -7.0, -9.85 m) [°C] | 17.09.2008-14.04.2014 |
| DL6 data logger | Soil moisture (-0.1, -0.2, -0.3, -0.4, -0.6, -1.0 m) [m$^3$ m$^{-3}$] | 01.07.2010-31.08.2014 |
| Frost probing | Active layer thaw depth [cm] | 01.09.2000-31.08.2014 |

Table 3: Physical and thermal subsurface properties used in model simulations.

| Material property | Units | Value |
|---|---|---|
| Residual saturation | - | 0 |
| Porosity | - | 0.4 |
| Permeability | m$^2$ | $10^{-12}$ |
| Density | kg m$^{-3}$ | 2650 |
| Heat capacity | J kg$^{-1}$ K$^{-1}$ | 850 |
| Thermal conductivity (saturated) | W m$^{-1}$ K$^{-1}$ | 1.7 |
| Thermal conductivity (dry) | W m$^{-1}$ K$^{-1}$ | 0.27 |





Table 4: Van Genuchten parameters α and m (Eq. 2) consistent with silt textures and as assigned in the seven model scenarios.

|  | Ref case | High alpha | Low alpha | High m | Low m | Max case | Min case |
|---|---|---|---|---|---|---|---|
| Van Genuchten α (Pa$^{-1}$) | 8·10$^{-4}$ | 12·10$^{-4}$ | 4·10$^{-4}$ | 8·10$^{-4}$ | 8·10$^{-4}$ | 12·10$^{-4}$ | 4·10$^{-4}$ |
| Van Genuchten m (-) | 0.19 | 0.19 | 0.19 | 0.29 | 0.10 | 0.29 | 0.10 |

Table 5: Active layer thickness (ALT) for the study period 2000-2014 and corresponding statistics, as observed at the UNISCALM grid (average of all grid points and also the centre location which is nearest the temperature logger) and as obtained from simulations.

| | Measured ALT (cm) UNISCALM site | | Modelled ALT (cm) Different soil moisture retention properties | | | | | | |
|---|---|---|---|---|---|---|---|---|---|
| Year | Grid average | Grid centre | Ref case | High alpha | Low alpha | High m | Low m | Max case | Min case |
| 2000 | 95 | 100 | 100 | 100 | 100 | 100 | 110 | 100 | 110 |
| 2001 | 99 | 102 | 100 | 110 | 110 | 110 | 120 | 100 | 130 |
| 2002 | 96 | 100 | 100 | 100 | 100 | 100 | 120 | 100 | 120 |
| 2003 | 93 | 99 | 100 | 100 | 100 | 100 | 110 | 90 | 110 |
| 2004 | 91 | 99 | 100 | 100 | 100 | 100 | 120 | 100 | 120 |
| 2005 | 74 | 79 | 100 | 100 | 100 | 90 | 110 | 90 | 110 |
| 2006 | 100 | 104 | 110 | 110 | 110 | 110 | 130 | 100 | 130 |
| 2007 | 105 | 106 | 100 | 100 | 100 | 100 | 110 | 100 | 110 |
| 2008 | 110 | 116 | 90 | 90 | 90 | 90 | 100 | 90 | 100 |
| 2009 | 100 | 106 | 100 | 100 | 100 | 100 | 120 | 100 | 120 |
| 2010 | 100 | 110 | 100 | 100 | 100 | 100 | 120 | 100 | 120 |
| 2011 | 101 | 104 | 110 | 110 | 110 | 100 | 120 | 100 | 120 |
| 2012 | 101 | 107 | 100 | 100 | 100 | 100 | 120 | 100 | 120 |
| 2013 | 103 | 111 | 110 | 110 | 110 | 110 | 130 | 110 | 130 |
| 2014 | 103 | 110 | 110 | 110 | 110 | 110 | 130 | 110 | 130 |
| **Statistics for time period 2000-2014** | | | | | | | | | |
| Min | 74 | 79 | 90 | 90 | 90 | 90 | 100 | 90 | 100 |
| Max | 110 | 116 | 110 | 110 | 110 | 110 | 130 | 110 | 130 |
| Mean | 98 | 104 | 102 | 103 | 103 | 101 | 118 | 99 | 119 |
| Std. dev. | 7.9 | 8.1 | 5.4 | 5.7 | 5.7 | 6.2 | 8.3 | 5.7 | 8.8 |
| Trend [cm yr$^{-1}$] | 0.8 | 1.0 | 0.5 | 0.3 | 0.3 | 0.3 | 0.8 | 0.6 | 0.6 |





Table 6: Subsurface temperature development [°C yr⁻¹] as observed and modelled (mean of all scenarios), as well as RMSE [°C] for all scenarios. Active layer trends down to 1.1 m depth are calculated for the period 09/2000-08/2014, and permafrost trends down to 9.85 m depth for the period 09/2008-08/2014. Summer is here defined as the period of continuous positive temperatures during thaw season and winter as the remaining part of the year.

| | | Trend [°C yr⁻¹] | | Root Mean Squared Error (RMSE) [°C] | | | | | | | |
| --- | --- | --- | --- | --- | --- | --- | --- | --- | --- | --- | --- |
| | Depth | Observed | Modeled (scenario mean) | Ref case | High alpha case | Low alpha case | High m case | Low m case | Max case | Min case | Mean |
| **Active layer** | 0.0 (summer) | -0.05 | n.a. | n.a. | n.a. | n.a. | n.a. | n.a. | n.a. | n.a. | **n.a.** |
| | 0.0 (winter) | 0.25 | n.a. | n.a. | n.a. | n.a. | n.a. | n.a. | n.a. | n.a. | **n.a.** |
| | 0.10 | 0.23 | 0.25 | 0.13 | 0.14 | 0.11 | 0.13 | 0.16 | 0.18 | 0.15 | **0.14** |
| | 0.20 | 0.26 | 0.24 | 0.09 | 0.06 | 0.13 | 0.07 | 0.04 | 0.01 | 0.05 | **0.06** |
| | 0.50 | 0.19 | 0.20 | 0.09 | 0.04 | 0.17 | 0.02 | 0.01 | 0.09 | 0.02 | **0.06** |
| | 1.10 | 0.17 | 0.16 | 0.26 | 0.21 | 0.33 | 0.14 | 0.18 | 0.02 | 0.21 | **0.19** |
| | **Mean for active layer** | | | **0.14** | **0.11** | **0.18** | **0.09** | **0.10** | **0.07** | **0.11** | |
| **Permafrost** | 2.00 | 0.14 | 0.13 | 0.22 | 0.18 | 0.28 | 0.12 | 0.18 | 0.03 | 0.20 | **0.17** |
| | 3.00 | 0.15 | 0.11 | 0.01 | 0.02 | 0.07 | 0.08 | 0.01 | 0.16 | 0.01 | **0.05** |
| | 5.00 | 0.11 | 0.09 | 0.03 | 0.00 | 0.07 | 0.05 | 0.03 | 0.11 | 0.05 | **0.05** |
| | 7.00 | 0.07 | 0.07 | 0.02 | 0.00 | 0.05 | 0.03 | 0.03 | 0.07 | 0.04 | **0.03** |
| | 9.85 | 0.05 | 0.05 | 0.03 | 0.03 | 0.02 | 0.03 | 0.03 | 0.03 | 0.02 | **0.03** |
| | **Mean for permafrost** | | | **0.06** | **0.05** | **0.10** | **0.06** | **0.06** | **0.08** | **0.06** | |
| | **Mean for entire depth** | | | **0.10** | **0.08** | **0.14** | **0.08** | **0.07** | **0.08** | **0.08** | |





Table 7: Simulated thaw progression to 1 m depth and active layer freeze-back in early winter 2011.

| Simulation scenario | Time to thaw 1 m depth [days] | Mean thaw rate [cm d⁻¹] | VoL. ice content [-] | Vol. unfrozen water content [-] | Vol. air content [-] | Effective therm. cond. $K_e$ [J kg⁻¹ K⁻¹] | Total latent heat [MJ] |
|---|---|---|---|---|---|---|---|
| **Ref case** | 75 | 1.3 | 0.25 | 0.05 | 0.10 | 0.58 | 33.6 |
| **High alpha** | 74 | 1.4 | 0.22 | 0.04 | 0.14 | 0.50 | 28.7 |
| **Low alpha** | 76 | 1.3 | 0.32 | 0.06 | 0.03 | 0.73 | 42.3 |
| **High m** | 78 | 1.3 | 0.21 | 0.01 | 0.18 | 0.47 | 28.1 |
| **Low m** | 58 | 1.7 | 0.21 | 0.14 | 0.05 | 0.54 | 27.4 |
| **Max case** | 85 | 1.2 | 0.15 | 0.01 | 0.24 | 0.34 | 20.2 |
| **Min case** | 60 | 1.7 | 0.23 | 0.16 | 0.02 | 0.60 | 30.8 |