# Peer review of "Soil moisture redistribution and its effect on inter-annual active layer temperature and thickness variations in a dry loess terrace in Adventdalen, Svalbard"

_The Cryosphere, 2016_

## Referee Comment (RC1) · Anonymous Referee #1 · 10 Sep 2016

The work presents a very simple endpoint sensitivity analysis of van Genuchten parameters and therefore soil water retention affect subsurface thermal hydrology, with specific attention paid to ice re-distribution due to cryosuction and unsaturated hydrology. The work is very specific to a dry site, and therefore has limited broad application to other sites within the pan-Arctic region. Similar and more extensive studies of subsurface hydro thermal parameters have been conducted previously, but to my knowledge few if any have been done in 'dry sites' and which compare results to more than just observed subsurface temperature, namely the inclusion of soil moisture, which is absolutely necessary when assessing the sensitivity of van Genuchten parameters.

[Figure]

I believe this inclusion of field data with the in depth modeling exercise produced some valuable insight into unsaturated thermal hydrology, which may prove valuable to the cryosphere community if the authors are able to focus both in the introduction and discussion of the need to quantify water retentions properties. The work is generally sound and free of technical errors and the authors do a fairly good job of making appropriate conclusions given the constraints of the modeling approach. Furthermore, the writing is clear and grammatically correct though not very concise or focused.

While I believe that this work will eventually achieve full publication I recommend that authors consider revising the manuscript to clearly state assumptions made in the modeling application, which have implications with regards to the interpretation of the results, though not necessarily problematic implications in my view. Furthermore, given the simplicity of the modeling exercise and the narrow scope of only perturbing two parameters within the van Genuchten equation, I believe it is vitally important to clearly motivate within the introduction why understanding water retentions regimes in permafrost systems is necessary. In this version of the manuscript, the introduction is unfocused and instead reads like a history of what research has been done regarding permafrost without much attempt to link it to soil moisture redistribution.

Major comments: 1) I therefore assume that there is no prescribed or simulated water fluxes in and out of the model domain, though it is not explicitly stated. While I see no huge reason why this would affect the validity of most of the results presented here, it should be remembered that any interpolation of the seasonality of the results should be taken with a grain of salt. In the results section the authors do an admirable job of pointing out when the model set-up without water fluxes in or out of the domain is responsible for deviations from observations. However, it maybe good in be supper clear about this set-up and state that what the boundary conditions of the model is. Particularly, that there is a no (water) flow in and out of the domain. In some documented cases water fluxes in and out as well as the shifted water retention location can have big consequences on the thermal regime of the subsurface i.e. [Atchley et al., 2016;

Helbig et al., 2013; McKenzie and Voss, 2013; Sjöberg et al., 2016].

Atchley, A. L., E. T. Coon, S. L. Painter, D. R. Harp, and C. J. Wilson (2016), Influences and interactions of inundation, peat, and snow on active layer thickness, Geophysical Research Letters. doi: 10.1002/2016GL068550.

Helbig, M., J. Boike, M. Langer, P. Schreiber, B. R. Runkle, and L. Kutzbach (2013), Spatial and seasonal variability of polygonal tundra water balance: Lena River Delta, northern Siberia (Russia), Hydrogeology Journal, 21(1), 133-147. doi: 10.1007/s10040-012-0933-4.

McKenzie, J. M., and C. I. Voss (2013), Permafrost thaw in a nested groundwater-flow system, Hydrogeology Journal, 21(1), 299-316. doi: 10.1007/s10040-012-0942-3.

Sjöberg, Y., E. Coon, K. Sannel, A. Britta, R. Pannetier, D. Harp, A. Frampton, S. L. Painter, and S. W. Lyon (2016), Thermal effects of groundwater flow through subarctic fens: A case study based on field observations and numerical modeling, Water Resources Research. doi: 10.1002/2015WR017571.

2) The work presents mainly endpoint and mid point evaluations of parameter space. While this type of exercise provides some insight into how parameters effect model output, there is no information about the middle parameter space and any non-linearity raising from combinations of van Genuchten parameters is hidden or lost. I would suggest that the authors attempt to simulate or at the very least discuss how combinations of van Genuchten parameters between those that are tested might behave. Could there be non-linarites as a result of untested combinations of van Genuchten parameters that lie with in the range of parameters tested?

3) It seems the central focus of the paper is how does unsaturated soil moisture distribution in the ALT and near surface permafrost layer affect the subsurface thermal regime at this relatively dry site. The introduction on the other hand reads like a history or what has been done, but it is my preference to use that history to highlight

why answering the unsaturated soil moisture distribution effect is important. This usually helps focus the paper and reader to why the results matter and produce a more precise manuscript.

Minor comments: Page 3 L 14-15: "Soil water retention is a critical, but highly uncertain parameter" I agree with this statement, and I believe the available literature also has evidence that supports this statement. Unfortunately, and despite the extensive literature cited in the introduction, the case that soil water retentions is critical, has not been made within the introduction of this paper, and therefore this statement and the purpose of the paper seems out to come out of no where. I suggest reshaping the introduction to be less of a history of what has been done to how the existing literature suggests that soil water retention may be important.

Page 5 Line: 12-13: "Active layer thickness was considered both for the grid centre, which is the point nearest the location of the ground temperature measurements, as well as for the average of all grid points" This is an awkward sentence. Do you mean ALT was measured at the grid center points and then averaged across an array of grid center points? I only see one observed time series in the figures, is this the average across the site?

Page 5 Line: 30: The unsaturated version of Darcy's law is Richard's equation.

Page 6 Line 31: Omit 'Then' in "Then different...."

Page 7 Line 2: It should be noted that setting residual saturation to zero in all cases 1) may produce the largest change in soil water content as all the water can drain out in dry cases, and 2) this formulation will allow all the pore water to go to ice during the winter, which will increase the winter thermal conductivity compared to systems where some pore water remains in a liquid state. Even thought the authors rightly point out that this assumption is often made it may still be worthwhile discussing these result in comparison to other more complete subsurface sensitivity studies such as [Harp et al., 2015], which includes residual saturation.

Harp, D., A. L. Atchley, S. L. Painter, E. Coon, C. Wilson, V. Romanovsky, and J. Rowland (2015), Effect of soil property uncertainties on permafrost thaw projections: a calibration-constrained analysis, The Cryosphere 10(3), 1-18. doi: 10.5194/tc-10-1-2016.

Page 7 Line 5: Omit 'Then' in "Then, both alpha . . ."

Page 7 Lines 5-10: I think this can be rephrased to be more clear and concise. Also, why were only 7 parameter combinations explored? Even though endpoint combinations can provide a lot of information about the behavior and sensitivity of parameters, there is little information about the model response to multiple combinations of parameters. Specifically any nonlinearities within the parameters space remain unknown.

Page 7 Lines 11-14: I think this needs rephrasing to be clearer, I would suggest something like, "Given that each combination of van Genuchten paramters will result in different soil moisture profiles under frozen conditions, each simulation test case with unique van Genuchten parameter combinations was spun-up and froze to attain unique ice-liquid-gas states"

Page 7 Lines 23-27: This provides reasoning in this modeling experiment to neglect water fluxes in and out of the domain. However the approach to neglect water fluxes is not clearly stated. While this is a huge simplification of the system I am ok with the approach, as long as it is clearly stated that a no water flux boundary is assigned. Please clearly state this boundary condition. Second, without the model able to represent transient water flows during the spin-up how can it be assured that the model is correctly representing the approximate amount of water in the system? Could this approach cause the mismatch between the observed and simulated water content in Figure 4? The reason I am ok with this approach here is that later the authors point out in the results when the model is unable to match observations. Which in my opinion highlights when representing a flux of water in and out of the system is necessary and when it is not, even for relatively dry sites, and thus becomes somewhat of a highlighted result in my opinion. This then begs the question, how much more important would representing surface and subsurface water flows in wet or highly transient sites be? Furthermore, given that van genuchten parameters were somewhat insensitive to subsurface temperatures in this study, would they be in sights the experience more transient hydrology?

Page 11 Lines 14-26: Though somewhat addressed in the next section (5.2), it maybe beneficial to discuss why the vertical movement or spreading of the ice thermal mass is important. I can invasion scenarios that create sharp or diffuse thermal gradients in the subsurface due to where and how concentrated the ice is.

Page 12 Lines 21-25: It would be interesting to extend the effective thermal conductivity evaluation to include differences in the location of ice mass in the subsurface, specifically compare the striated ice distribution (Fig 3, b) to the diffuse distribution (Fig 3. C). Does the striation of ice change effective thermal conductivity?

Page 13 Section 5.3: I appreciate this discussion that addresses ALT characteristics beyond the scope of soil moisture distribution and how seasonal differences i.e. winter versus summer, have been shown in literature and the present study to act differently on ALT. However, I think it too should be discussed within the context of soil moisture distribution. While in general it may be counter intuitive that ALT is more responsive to winter conditions then summer, but for those of us working on permafrost it makes since. In the Arctic winters are long, summers are short and the ground is mostly in a frozen state. Furthermore, ice is more thermally conductive than water and therefore a cold signal or lack thereof in the winter will propagate further into the subsurface. Given that winter conditions are important, this work should then address how does soil moisture distribution and therefore ice distribution in the winter moderate the winter time signal. Does it at all? If so, how does it? Given that this experiment is in a dry site with little water moving through the subsurface, can the conclusions be applied to wet sites with lots of subsurface flow? What further research would be necessary to answer these issues?

---

## Referee Comment (RC2) · Anonymous Referee #2 · 18 Sep 2016

Review of manuscript tc-2016-173 "Soil moisture resdistribution and its effect on inter-annual active layer temperature and thickness variations in a dry loess terrace in Adventdalen, Svalbard" by C. Schuh et al.

General comments:
This manuscript investigates freeze/thaw dynamics in a soil profile for a 14-years time series of measured data from the UNISCALM-site on Svalbard with the aid of a numerical model. Specifically, for a homogeneous silt profile, the van Genuchten parameters $\alpha$ and $n$ are varied in a reasonable range. Differences in thaw depth, water and ice

content are interpreted i) for a quasi-synthetic test case using upper and lower boundary conditions measured in the field and ii) compared to field observations. The paper is very well written and fits well into the scope of The Cryosphere. I have one major concern which is the fit between measured and modeled data which – in my opinion - needs major revision or restructuring of the paper before the manuscript can be recommended for publication.

Major comments: Run in a quasi-synthetic mode, the model is very helpful for exploring the effects of variations in van Genuchten $\alpha$ and $n$ on thaw depth as well as water and ice distribution throughout a silty soil profile (cf. Fig. 3). In this case, a rather simple test case is generated where modelled data depend only on the chosen parameterizations of the soil profile and the imposed upper and lower boundary conditions. With these simulations, processes can be interpreted based on the assumed conceptual model without real linkage to field observations and this is done very well in this study. However, as soon as simulations are compared to measured field data, especially Figure 4 shows that there are still large discrepancies between modelled data and observations and the model is not yet able to reproduce freeze/thaw processes observed in the field. For example, it is definitely not sufficient when summer data at one depth of the profile fit to summer simulations of one test case and winter data at the same depth of the profile fit to winter data of another test case. Here, the challenge is to set up a conceptual model and to find a parameterization that is able to reproduce observations (temperature, moisture, ice content) at all depths during the complete time series before processes occurring at the site can be interpreted and quantified safely. Finding such a paramterization could be quite some effort, so probably it is the better choice for this paper to reduce the study to the synthetic cases and remove the sections comparing measured and modelled data. The alternative would be to "calibrate" the model such that simulations are able to reproduce the field observations.

Specific comments:

P 1, L 28: correct "temperatures"

P 2, L 26: Which controlling factors? Please add related information.

P 3, L 18-21: The two specific aims are very closely related. Please reformulate the major aims of the study.

P 4, L 29: correct "100 m x 100" m or "100 x 100 m$^2$"

P 5, L 10: please add probe to Table 2 P 6, L 28: The vertical resolution of the model (0.1 m) is rather coarse. Especially, close to the ground surface, resolutions of 0.01 m or even less are often required to adequately reproduce temperature and moisture gradients. Did the authors check the performance of the model in this regard?

P 6, L 31: please add reference for the chosen parameter set

P7, L 9-14: Please clarify initial condition: As far as I understand, capillary pressure was linearly interpolated with 0 hPa at 1.2 m depth and -120 hPa at ground surface?

P 7, L 21: please correct: linearly

P 8, Sect. 4.1: Table 5 is not very well suited for comparing measured and modelled data. A plot like Figure 4 would be much more helpful for assessing the quality of the different models.

P 9, L- 13: correct "system"

P 9, Section 4.2: Case studies discussed in the text and shown in Fig. 4 are not the same. Simulations shown in Fig. 4 do not reproduce measured values.

Sect. 5: The general discussion of the influence of $\alpha$ and $n$ on the processes occurring in the soil profile is well done and okay as long as it is based on the synthetic cases.

---

## Author Comment (AC1) · 16 Nov 2016

**Response to Anonymous Referee 1**

*Referee comments shown as "RC:", author replies as "AR:".*

RC: The work presents a very simple endpoint sensitivity analysis of van Genuchten parameters and therefore soil water retention affect subsurface thermal hydrology, with specific attention paid to ice re-distribution due to cryosuction and unsaturated hydrology. The work is very specific to a dry site, and therefore has limited broad application to other sites within the pan-Arctic region. Similar and more extensive studies of subsurface hydro thermal parameters have been conducted previously, but to my knowledge few if any have been done in 'dry sites' and which compare results to more than just observed subsurface temperature, namely the inclusion of soil moisture, which is absolutely necessary when assessing the sensitivity of van Genuchten parameters.

RC: I believe this inclusion of field data with the in depth modeling exercise produced some valuable insight into unsaturated thermal hydrology, which may prove valuable to the cryosphere community if the authors are able to focus both in the introduction and discussion of the need to quantify water retentions properties. The work is generally sound and free of technical errors and the authors do a fairly good job of making appropriate conclusions given the constraints of the modeling approach. Furthermore, the writing is clear and grammatically correct though not very concise or focused.

**RC #1:** While I believe that this work will eventually achieve full publication I recommend that authors consider revising the manuscript to clearly state assumptions made in the modeling application, which have implications with regards to the interpretation of the results, though not necessarily problematic implications in my view. Furthermore, given the simplicity of the modeling exercise and the narrow scope of only perturbing two parameters within the van Genuchten equation, I believe it is vitally important to clearly motivate within the introduction why understanding water retentions regimes in permafrost systems is necessary. In this version of the manuscript, the introduction is unfocused and instead reads like a history of what research has been done regarding permafrost without much attempt to link it to soil moisture redistribution.

*AR #1: We thank Anonymous Referee 1 for the valuable suggestions to improve and strengthen our paper; the text has been thoroughly revised and all comments have been addressed in our responses below. We acted on your suggestion and condensed the introduction section to better carve out the particular aim of this study. Also, we now stress the importance of soil water retention properties and the resulting moisture distribution more clearly when discussing thaw progression and active layer thickness variations over time.*

**Major comments:**

**RC #2 MAJOR COMMENT-1:** I therefore assume that there is no prescribed or simulated water fluxes in and out of the model domain, though it is not explicitly stated. While I see no huge reason why this would affect the validity of most of the results presented here, it should be remembered that any interpolation of the seasonality of the results should be taken with a grain of salt. In the results section the authors do an admirable job of pointing out when the model set-up without water fluxes in or out of the domain is responsible for deviations from observations. However, it maybe good in be supper clear about this set-up and state that what the boundary conditions of the model is. Particularly, that there is a no (water) flow in and out of the domain. In some documented cases water fluxes in and out as well as the shifted water retention location can have big consequences on the thermal regime of the subsurface i.e. [Atchley et al., 2016; Helbig et al., 2013; McKenzie and Voss, 2013; Sjöberg et al., 2016].

Atchley, A. L., E. T. Coon, S. L. Painter, D. R. Harp, and C. J. Wilson (2016), Influences and interactions of inundation, peat, and snow on active layer thickness, Geophysical Research Letters. doi: 10.1002/2016GL068550.

Helbig, M., J. Boike, M. Langer, P. Schreiber, B. R. Runkle, and L. Kutzbach (2013), Spatial and seasonal variability of polygonal tundra water balance: Lena River Delta, northern Siberia (Russia), Hydrogeology Journal, 21(1), 133-147. doi: 10.1007/s10040-012-0933-4.

McKenzie, J. M., and C. I. Voss (2013), Permafrost thaw in a nested groundwater-flow system, Hydrogeology Journal, 21(1), 299-316. doi: 10.1007/s10040-012-0942-3.

Sjöberg, Y., E. Coon, K. Sannel, A. Britta, R. Pannetier, D. Harp, A. Frampton, S. L. Painter, and S. W. Lyon (2016), Thermal effects of groundwater flow through subarctic fens: A case study based on field observations and numerical modeling, Water Resources Research. doi: 10.1002/2015WR017571.

*AR #2 MAJOR COMMENT-1: It is correct that we assume no water fluxes in and out of the system and therefore assign no-flow boundaries to the model domain. Advective heat transport by lateral water flow is likely to be important for many active layer systems and we now more clearly highlight this and refer to the relevant literature in our revised introduction section. However, at the very dry UNISCALM site, there is strong evidence that the system is largely unaffected by lateral water fluxes and very little infiltration occurs (see page 4, lines 19-24 of the revised text). This has been investigated in detail in previous studies [cited as Schuh, 2015], where summer rainfall and snowmelt infiltration derived from hydro-meteorological field site data was applied to the surface. In that study it was shown that the small amount of infiltration in comparison to the existing amount of ground/pore ice did not have a notable effect on the inter-annual ALT variation at that site. Section 3.3 of our manuscript has been revised to more clearly state the assigned boundary conditions (page 7 line 30 – page 8, line 2) as well as the motivation for deliberately neglecting water fluxes at this particular study site (page 8, lines 15-21): "Lateral water fluxes through and infiltration into the system were neglected and no flow boundaries assigned to all faces of the model domain. While recognizing that flowing water can have considerable thermal effects on active layer processes, its exclusion is considered an appropriate assumption for the dry UNISCALM site (cf. Sect. 2)."*

**RC #3 MAJOR COMMENT-2:** The work presents mainly endpoint and mid point evaluations of parameter space. While this type of exercise provides some insight into how parameters effect model output, there is no information about the middle parameter space and any non-linearity raising from combinations of van Genuchten parameters is hidden or lost. I would suggest that the authors attempt to simulate or at the very least discuss how combinations of van Genuchten parameters between those that are tested might behave. Could there be non-linarites as a result of untested combinations of van Genuchten parameters that lie within the range of parameters tested?

*AR #3 MAJOR COMMENT-2: This study is laid out as a scenario analysis which is why we used a combination of average parameter values as reference case, varied both parameters α and m independently by ±50%, and defined two more cases as the higher and lower end of this parameter space. This is a careful selection of parameters corresponding to a systematic range of values for the soil texture type observed at the UNISCALM site (silt loam) and consistent with information obtained from the UNSODA unsaturated soil hydraulic database as well as with the range of retention parameters reported in literature for that soil class. It is correct that by testing these seven cases, any other parameter combination within this range or potentially outside this range remains unconsidered. While this choice of representative scenarios should be sufficient for our study, we acknowledge the necessity to point out the general value of systematic parameter investigations due to the nonlinear nature of the retention curve. This is now highlighted in our revised text (section 3.3 of the revised manuscript; page 7, lines 27-30, as follows: "Given that each combination of van Genuchten parameters will result in different soil moisture profiles, each simulation test case was set-up to attain unique ice-liquid-gas states. Still, despite the systematic approach of testing relevant parameter combinations, note that any potential nonlinearities arising from untested combinations within the parameter space remain unconsidered in this analysis.")*

**RC #4 MAJOR COMMENT-3:** It seems the central focus of the paper is how does unsaturated soil moisture distribution in the ALT and near surface permafrost layer affect the subsurface thermal regime at this relatively dry site. The introduction on the other hand reads like a history or what has been done, but it is my preference to use that history to highlight why answering the unsaturated soil moisture distribution effect is important. This usually helps focus the paper and reader to why the results matter and produce a more precise manuscript.

*AR #4 MAJOR COMMENT-3: We acted on your suggestion and revised the introduction accordingly. We removed those parts that did not contribute directly to the main objective of this study, i.e. the more general review of permafrost-hydrological modeling studies and instead, we placed more focus on the relevance of soil moisture for active layer processes and its importance for climate change*

*investigations, especially in cold regions: "Both soil moisture and the thickness of the active layer have been identified as Essential Climate Variables (GCOS, 2015). Soil moisture is an important variable for energy exchange and governs the processes occurring at the land-atmosphere interface by partitioning incoming solar radiation into fluxes of latent and sensible heat (GCOS, 2015). Soil moisture controls subsurface physical properties such as thermal conductivity and heat capacity, so that the movement of moisture within the subsurface is essential for understanding the water and heat balance of the ground, particularly in cold climates (Wu et al., 2016). Still, most studies on permafrost thaw projections mainly investigate the structural uncertainty in climate models, whereas the parametric uncertainty in soil properties is barely accounted for (Harp et al., 2016)." (page 2, lines 12-18 of the revised manuscript).*

**Minor comments:**

**RC #5:** Page 3 L 14-15: "Soil water retention is a critical, but highly uncertain parameter" I agree with this statement, and I believe the available literature also has evidence that supports this statement. Unfortunately, and despite the extensive literature cited in the introduction, the case that soil water retentions is critical, has not been made within the introduction of this paper, and therefore this statement and the purpose of the paper seems out to come out of no where. I suggest reshaping the introduction to be less of a history of what has been done to how the existing literature suggests that soil water retention may be important.

*AR #5: This section has been revised substantially. Please refer to our response to MAJOR COMMENT-3 (AR #4).*

**RC #6:** Page 5 Line: 12-13: "Active layer thickness was considered both for the grid centre, which is the point nearest the location of the ground temperature measurements, as well as for the average of all grid points" This is an awkward sentence. Do you mean ALT was measured at the grid center points and then averaged across an array of grid center points? I only see one observed time series in the figures, is this the average across the site?

*AR #6: ALT was measured annually at all 121 nodes of the 100 m x 100 m UNISCALM grid (see page 5, lines 12-14). First, we consider the ALT measurements exclusively from the node in the grid center because it coincides with the location of the utilized subsurface temperature dataset (obtained from the Tinytag loggers, see page 5, lines 14-17). Second, we use the annual ALT average of all 121 grid nodes to reduce measurement uncertainty and obtain a more robust dataset. In our analysis, we refer to the ALT measurements as "grid average" and "grid center" (see Fig. 6a-b, Tab. 5 of the revised manuscript). In the manuscript, we clarified this paragraph by minor adjustments to section 3.1.*

**RC #7:** Page 5 Line: 30: The unsaturated version of Darcy's law is Richard's equation.

*AR #7: Yes, the model (ATS) couples RE with an equation for heat transport which accounts for latent heat transfer, where several constitutive relationships are used to close the system of equations, including Darcy's law and thermodynamic constraints. This is further clarified in the revised manuscript (page 6, lines 14-16).*

**RC #8:** Page 6 Line 31: Omit 'Then' in "Then different. . .."

*AR #8: Deleted: "Then".*

**RC #9:** Page 7 Line 2: It should be noted that setting residual saturation to zero in all cases 1) may produce the largest change in soil water content as all the water can drain out in dry cases, and 2) this formulation will allow all the pore water to go to ice during the winter, which will increase the winter thermal conductivity compared to systems where some pore water remains in a liquid state. Even thought the authors rightly point out that this assumption is often made it may still be worthwhile discussing these result in comparison to other more complete subsurface sensitivity studies such as [Harp et al., 2015], which includes residual saturation.

Harp, D., A. L. Atchley, S. L. Painter, E. Coon, C. Wilson, V. Romanovsky, and J. Rowland (2015), Effect of soil property uncertainties on permafrost thaw projections: a calibration-constrained analysis, The Cryosphere 10(3), 1-18. doi: 10.5194/tc-10-1- 2016.

*AR #9: We have updated the text and now include a highlight of our results in the context of the highly relevant study by Harp et al. (2016), as follows: "The missing sensitivity of soil water retention parameters to inter-annual ALT variation is not surprising. Also Harp et al. (2016) found the van Genuchten parameters not to affect subsurface temperatures in a long-term thaw projection study. This might be explained by the fact that in areas subject to a seasonal freeze/thaw cycle, the water retention parameters mainly control the seasonal soil moisture (re-)distribution during freezing and thaw (cf. 5.1). While we found the different retention curves to affect the rate of thaw progression and thus the respective annual thaw depth, the total period of active layer development from thaw to freeze-back was similar in all cases (cf. Sec. 5.2). Water retention characteristics therefore seem to be relevant mainly on a short time scale during thaw and freeze-up." (page 15, from line 30 onwards of the revised version).*

**RC #10:** Page 7 Line 5: Omit 'Then' in "Then, both alpha . . ."

*AR #10: The manuscript text has been clarified (page 7, lines 21-23).*

**RC #11:** Page 7 Lines 5-10: I think this can be rephrased to be more clear and concise. Also, why were only 7 parameter combinations explored? Even though endpoint combinations can provide a lot of information about the behavior and sensitivity of parameters, there is little information about the model response to multiple combinations of parameters. Specifically any nonlinearities within the parameters space remain unknown.

*AR #11: This part of the manuscript text has been re-written to make the presentation more concise (page 7, lines 21-26). For the choice of seven parameter combinations and potential nonlinearities of the parameter space please refer to our response to MAJOR COMMENT-2 (AR # 3)*

**RC #12:** Page 7 Lines 11-14: I think this needs rephrasing to be clearer, I would suggest something like, "Given that each combination of van Genuchten paramters will result in different soil moisture profiles under frozen conditions, each simulation test case with unique van Genuchten parameter combinations was spun-up and froze to attain unique ice liquid- gas states"

*AR #12: The section has been rephrased accordingly. We also included a clearer statement of the applied boundary conditions (page 7, line 27 ff).*

**RC #13:** Page 7 Lines 23-27: This provides reasoning in this modeling experiment to neglect water fluxes in and out of the domain. However the approach to neglect water fluxes is not clearly stated. While this is a huge simplification of the system I am ok with the approach, as long as it is clearly stated that a no water flux boundary is assigned. Please clearly state this boundary condition. Second, without the model able to represent transient water flows during the spin-up how can it be assured that the model is correctly representing the approximate amount of water in the system? Could this approach cause the mismatch between the observed and simulated water content in Figure 4? The reason I am ok with this approach here is that later the authors point out in the results when the model is unable to match observations. Which in my opinion highlights when representing a flux of water in and out of the system is necessary and when it is not, even for relatively dry sites, and thus becomes somewhat of a high-lighted result in my opinion. This then begs the question, how much more important would representing surface and subsurface water flows in wet or highly transient sites be? Furthermore, given that van genuchten parameters were somewhat insensitive to subsurface temperatures in this study, would they be in sights the experience more transient hydrology?

*AR #13: This section was reformulated and is now more precise in stating the applied boundary conditions. For the assumptions made on water fluxes through the system please refer to our response to MAJOR COMMENT-1 (AR #2). Regarding the correct amount of water in the closed*

*system, we generally represent field site conditions by saturating the unfrozen model domain below about 2 m depth. Assigning a pressure of -200 hPa as the top boundary condition before model freezing and spin-up resulted in a water content in the top cell of 0.11-0.32 (depending on scenario) and a linear increase up to full saturation (0.4) in the model domain below about 2 m depth. This is consistent with the water content both in the permafrost (0.4) and in the active layer (0.1-0.25) recorded at the field site. Please note that after freezing the model domain, each simulation test case was subject to a transient spin-up (allowing water to redistribute in the subsurface) until a periodic steady state was achieved (page 8, lines 6-9). Furthermore, previous modeling studies by Schuh [2015] (already cited) showed that infiltration estimated from meteorological field site data only accounted for an insignificant fraction of the existing water content in the system. Therefore, it seems unlikely that the mismatch in water content seen between field site data and some of the model cases is solely caused by the assumption of no water fluxes in to the system.*

**RC #14:** Page 11 Lines 14-26: Though somewhat addressed in the next section (5.2), it maybe beneficial to discuss why the vertical movement or spreading of the ice thermal mass is important. I can invasion scenarios that create sharp or diffuse thermal gradients in the subsurface due to where and how concentrated the ice is.

**AR #14**: *The manuscript was substantially revised with regard to the particular effect of ice striation on thermo-hydrological subsurface processes e.g. for the progression of the thawing front (please see our response AR#15 to comment RC #15 below). We also included a short general discussion on the relevance of the vertical spreading of the ice in section 5.3 (page 15, lines 30ff).*

**RC #15:** Page 12 Lines 21-25: It would be interesting to extend the effective thermal conductivity evaluation to include differences in the location of ice mass in the subsurface, specifically compare the striated ice distribution (Fig 3, b) to the diffuse distribution (Fig 3. C). Does the striation of ice change effective thermal conductivity?

**AR #15:** *We extended our analysis on thaw progression in section 5.2 insofar as we now also consider the local rate of thaw within the soil profile in addition to the integrated consideration of the 1 m soil column (Tab. 7). We use the scenarios shown in Fig. 3 with different striation to illustrate the different rates of thaw in systems characterized by different soil moisture contents and distribution, ranging from diffuse to highly striated ice distribution. Please see the revised manuscript and the new Fig. 5 (relevant excerpts of the text reproduced in the following).*

*"Furthermore, model simulations show that not only the amount of soil moisture, but also the distribution of ice within the active layer impacts the progression of the thawing front. Fig. 5 compares the ice content profile before the onset of thaw in 2011 to the respective thaw rate when the ground surface temperatures become positive. As shown previously (cf. Fig. 3), model scenarios 'ref case' (black), 'high m' (green) and 'max case' (orange) resulted in a distinct stratification of ice, whereas 'low m' (blue) shows almost a linear increase in ice content towards the permafrost table (Fig. 5a). In all scenarios, thawing the upper 10 cm of the ground occurs at a relatively low rate between 1.3-1.4 cm/d (Fig. 5b) mainly because air temperatures are still low, but also because of latent heat consumption due to the increased ice content in the thin layer just beneath the surface. Once the thawing front passes the ice layer at 5 cm depth, the thaw rate increases to its maximum of 5 cm/d, fuelled by markedly increasing air temperatures (not shown) and regardless of the respective soil moisture content. Below 20 cm depth, all scenarios show a generally decreasing thaw rate, but with notably case-specific differences only between 40 cm and 60 cm depth. Here, discrepancies originate from the particular ice-liquid-gas composition in the thawing ground and its effect on heat propagation, as discussed above. Furthermore, enhanced thaw due to advective heat transport through previously frozen water is likely to occur in more saturated systems such as 'ref case' (e.g. at 50 cm depth). Scenarios 'high m' and 'low m' show discrepancies in their thaw rate, while having comparable ice contents. In 'high m', the thaw rate drops to 1.4 cm/d when reaching the clearly defined ice-rich zone at 60 cm depth, whereas the thaw rate at that depth is much higher (2.5 cm/d) in the 'low m' case, characterized by a more stable ice content profile." (page 14, lines 7-23)*

[Figure]

*Figure 5: Comparison of (a) simulated ice content distribution with depth at the end of winter 2011 (01.05.2011) and (b) the corresponding modelled rate of active layer thaw starting after the first day of positive ground surface temperatures (25.05.2011), exemplified by four selected model scenarios.*

**RC #16:** Page 13 Section 5.3: I appreciate this discussion that addresses ALT characteristics beyond the scope of soil moisture distribution and how seasonal differences i.e. winter versus summer, have been shown in literature and the present study to act differently on ALT. However, I think it too should be discussed within the context of soil moisture distribution. While in general it may be counter intuitive that ALT is more responsive to winter conditions then summer, but for those of us working on permafrost it makes since. In the Arctic winters are long, summers are short and the ground is mostly in a frozen state. Furthermore, ice is more thermally conductive than water and therefore a cold signal or lack thereof in the winter will propagate further into the subsurface. Given that winter conditions are important, this work should then address how does soil moisture distribution and therefore ice distribution in the winter moderate the winter time signal. Does it at all? If so, how does it? Given that this experiment is in a dry site with little water moving through the subsurface, can the conclusions be applied to wet sites with lots of subsurface flow? What further research would be necessary to answer these issues?

*AR #16: We appreciate your suggestion. Section 5.3 has now been extended by a more detailed paragraph about the role of the particular water retention curve for inter-annual ALT variation. This was an important step also to synthesize our findings and conclusions from the previous sections regarding the sensitivity of retention parameters to seasonal vs. inter-annual active layer dynamics. This way we believe to have polished the manuscript further to make in more focused on the importance of soil moisture (re-)distribution. Nonetheless, this comment certainly highlights a need for further research, in particular considering how sites with different hydro-climatic conditions may yield different active layer moisture/wetness conditions and ice dynamics.*

*"The missing sensitivity of soil water retention parameters to inter-annual ALT variation is not surprising. Also Harp et al. (2016) found the van Genuchten parameters not to affect subsurface temperatures in a long-term thaw projection study. This might be explained by the fact that in areas*

*subject to a seasonal freeze/thaw cycle, the water retention parameters mainly control the seasonal soil moisture (re-)distribution during freezing and thaw (cf. 5.1). While we found the different retention curves to affect the rate of thaw progression and thus the respective annual thaw depth, the total period of active layer development from thaw to freeze-back was similar in all cases (cf. Sec. 5.2). Water retention characteristics therefore seem to be relevant mainly on a short time scale during thaw and freeze-up." (page 16, lines 30ff).*

---

## Author Comment (AC2) · 16 Nov 2016

**Response to Anonymous Referee 2**

*Referee comments shown as "RC:", author replies as "AR:".*

General comments:

**RC #1 GENERAL COMMENT:** This manuscript investigates freeze/thaw dynamics in a soil profile for a 14-years time series of measured data from the UNISCALM-site on Svalbard with the aid of a numerical model. Specifically, for a homogeneous silt profile, the van Genuchten parameters α and n are varied in a reasonable range. Differences in thaw depth, water and ice content are interpreted i) for a quasi-synthetic test case using upper and lower boundary conditions measured in the field and ii) compared to field observations. The paper is very well written and fits well into the scope of The Cryosphere. I have one major concern which is the fit between measured and modeled data which – in my opinion - needs major revision or restructuring of the paper before the manuscript can be recommended for publication.

*AR # 1 GENERAL COMMENT: We thank Anonymous Referee 2 for valuable suggestions to improve and strengthen our paper; all comments have been thoroughly addressed in our responses below. With regard to the major concern regarding the comparability of model results and field site data, we revised the manuscript insofar as we removed text/paragraphs focusing on detailed quantitative comparison. Also, we rephrased the introduction and study aim to distinguish more clearly the character and claims of our work from that of a calibration study.*

Major comments:

**RC #2 MAJOR COMMENT:** Run in a quasi-synthetic mode, the model is very helpful for exploring the effects of variations in van Genuchten α and n on thaw depth as well as water and ice distribution throughout a silty soil profile (cf. Fig. 3). In this case, a rather simple test case is generated where modelled data depend only on the chosen parameterizations of the soil profile and the imposed upper and lower boundary conditions. With these simulations, processes can be interpreted based on the assumed conceptual model without real linkage to field observations and this is done very well in this study. However, as soon as simulations are compared to measured field data, especially Figure 4 shows that there are still large discrepancies between modelled data and observations and the model is not yet able to reproduce freeze/thaw processes observed in the field. For example, it is definitely not sufficient when summer data at one depth of the profile fit to summer simulations of one test case and winter data at the same depth of the profile fit to winter data of another test case. Here, the challenge is to set up a conceptual model and to find a parameterization that is able to reproduce observations (temperature, moisture, ice content) at all depths during the complete time series before processes occurring at the site can be interpreted and quantified safely. Finding such a paramterization could be quite some effort, so probably it is the better choice for this paper to reduce the study to the synthetic cases and remove the sections comparing measured and modelled data. The alternative would be to "calibrate" the model such that simulations are able to reproduce the field observations.

*AR #2 MAJOR COMMENT: It is not our intention or aim to conduct a model calibration; rather our general main objective is to investigate effects of different soil water retention properties on active layer dynamics. This site is chosen on the basis of previous initial investigations (Schuh, 2015) showing that the site is very dry and unsaturated and potentially highly influenced by cryosuction effects. Therefore in this study we conduct a scenario analysis where we investigate different van Genuchten parameter combinations applicable for the site conditions with the objective to improve the understanding of the physical processes governing the dynamics of an unsaturated active layer as found at/consistent with the UNISCALM site. We derive our simulation test cases from field information (temperature and pressure boundary conditions and sediment properties) and again use field site data (ALT and water content measurements as well as cryostratigraphic information) to place the analysis and model results in the context of this particular site. The comparison between field site observations*

*and model results is done to classify the different scenarios with regard to certain field site characteristics, for example the measured water content at the field site (Fig. 4) is compared to simulations not only as an indicator for the correct amount of water in the system, but also to derive information on thaw progression.*

*We revised the manuscript insofar as to avoid misinterpretations of our intentions with the study and the quantitative comparisons, starting by the introduction and a clearer statement of the purpose of our study (page 3, lines 30ff). We also eliminated the quantification of root mean squared errors (RMSE) for the differences between simulated and measured ground temperatures in section 3.3 (page 8, lines 30ff) and section 4.1 (page 9 lines 20) to avoid implications of a calibration study.*

**Specific comments:**

**RC #3:** P 1, L 28: correct "temperatures"

*AR #3: Corrected: "temperatures".*

**RC #4:** P 2, L 26: Which controlling factors? Please add related information.

*AR #4: Information was added: "[…] key controlling factors of active layer development, mainly air temperature and solar radiation, […]."*

**RC #5:** P 3, L 18-21: The two specific aims are very closely related. Please reformulate the major aims of the study.

*AR #5: The objective of the study has been revised and reformulated as follows: "The aim is to study how soil moisture retention properties affect moisture and ice (re-)distribution as well as subsurface temperature and active layer thickness variations in the partially saturated active layer under multiple freeze-thaw cycles. In a scenario analysis approach, the different soil moisture retention properties are expressed through careful selection of relevant parameter values derived from field information, and simulation results are put in the context of the particular UNISCALM study site and other relevant permafrost environments." (page 4, lines 22ff).*

**RC #6:** P 4, L 29: correct "100 m x 100" m or "100 x 100 m2"

*AR #6: Corrected: "100 m x 100 m".*

**RC #7:** P 5, L 10: please add probe to Table 2

*AR #7: We previously stated the probe as ""DL6 Data Logger" in Table 2. This notation was misleading; we now changed it to "Delta-T profile probe" to comply with the text.*

**RC #8:** P 6, L 28: The vertical resolution of the model (0.1 m) is rather coarse. Especially, close to the ground surface, resolutions of 0.01 m or even less are often required to adequately reproduce temperature and moisture gradients. Did the authors check the performance of the model in this regard?

*AR #8: Numerical convergence is assured by careful selection of convergence criteria, combined with the use of robust numerical computation routines (for details we refer to Painter et al., 2016, which is now also cited in our revised version). The mesh resolution is selected based on model needs and intention/purpose of the investigation performed; here a mesh of 0.1 m is deemed sufficient since we focus on general active layer dynamics for homogeneous soil texture and using ground surface temperature as thermal boundary condition. As such, surface heat attenuation processes (snow cover, ponding, vegetation, etc) which otherwise may require more careful consideration of near-surface and surface mesh discretization are avoided.*

*Painter, S.L., Coon, E.T., Atchley, A.L., Berndt, M., Garimella, R., Moulton, J.D., Svyatskiy, D., Wilson, C.J., 2016. Integrated surface/subsurface permafrost thermal hydrology: Model formulation and proof-of-concept simulations. Water Resources Research. doi:10.1002/2015WR018427*

**RC #9:** P 6, L 31: please add reference for the chosen parameter set

*AR #9: References were added to Table 3:*

*Andersland, O.B. and Ladanyi, B. (1994): An introduction to frozen ground engineering. Dordrecht (Springer), ISBN: 978-1-4757-2290-1.*

*Fitts, C. (2013): Groundwater science. 2nd edition, Oxford (Elsevier), doi: 10.1016/B978-0-12-384705-8.00016-9.*

*Freeze, R.A. and Cherry, J.A. (1979): Groundwater. Hemel Hempstead (Prentice), ISBN: 978-0133653120.*

*Huang, P.M.; Li, Y. and Sumner, M.E. (eds.) (2012): Handbook of soil sciences – properties and processes. 2nd edition, Boca Raton (Taylor & Francis), ISBN: 978-1-4398-0305-9.*

*Kirsch, R. and Yaramanci, U. (2006): Geophysical characterization of aquifers. In: Kirsch, R.(ed.), Groundwater geophysics - a tool for hydrology. Berlin (Springer), pp. 439-457,ISBN: 978-3-540-29383-5.*

*Ochsner, T.E.; Horton, R. and Ren, T. (2001): A new perspective on soil thermal properties. Soil Sci. Soc. Am. J. 65, pp. 1641–1647.*

*Schwartz, F.W. and Zhang, H. (2003): Fundamentals of ground water. New York (Wiley),ISBN: 978-0-471-13785-6.*

*Wesley, L.D. (2010): Fundamentals of soil mechanics for sedimentary and residual soils. Hoboken (Wiley), ISBN: 978-0-470-37626-3.*

**RC #10:** P7, L 9-14: Please clarify initial condition: As far as I understand, capillary pressure was linearly interpolated with 0 hPa at 1.2 m depth and -120 hPa at ground surface?

*AR #10: We first put the water table at about -2 m by assigning a pressure of about -200 hPa as top boundary condition and then interpolating linearly with depth. Then we froze the model domain, resulting in the water table to move up to about -1.2 m. We clarified this in the manuscript by reformulating the describing the model setup (page 7, lines 27ff).*

**RC #11:** P 7, L 21: please correct: linearly

*AR #11: Done.*

**RC #12:** P 8, Sect. 4.1: Table 5 is not very well suited for comparing measured and modeled data. A plot like Figure 4 would be much more helpful for assessing the quality of the different models.

*AR #12: We took into consideration to display a graph of selected data only and to move Table 5 to the supplementary materials. Eventually we decided to keep Table 5 in the text. We believe that in the results section covering the differences in ALT with regard to certain retention properties it is important to show the complete findings, i.e. the two field site datasets and all seven model scenarios, including their statistical characteristics. This is not practicable in a plot due to the large number of simulation cases which obfuscates comparison. Also, since we would like to avoid the direct comparison between field observations and model results (see our response to major comment above [AR # 2]), we feel that the table is more suitable.*

**RC #13:** P 9, L- 13: correct "system"

*AR #13: Done.*

**RC #14:** P 9, Section 4.2: Case studies discussed in the text and shown in Fig. 4 are not the same. Simulations shown in Fig. 4 do not reproduce measured values.

*AR #14: The discussion regarding soil moisture development (Figure 4) has been revised and is now focused exclusively on those three scenarios shown in Figure 4. We also added a note to the caption of Figure 4 stating that "The remaining simulation cases (not shown) reside within the limits of the min and max cases." About the match between modeled and observed soil moisture please refer to our response to the major comment above (AR #2).*

**RC #15:** Sect. 5: The general discussion of the influence of α and n on the processes occurring in the soil profile is well done and okay as long as it is based on the synthetic cases.

*AR #15: Please refer to our response to the major comment concerning the comparison of simulations to field data (AR #2).*

---

## Author Response (AR2)

**Author Response letter**

*We thank the Editor and the two Anonymous Referees for reviewing the second submission of our manuscript and for providing constructive and helpful suggestions for improvements. We have addressed the comments and provide our responses below. Editor comments are shown as EC, Referee comments as RC, and author responses as AR. Page and line numbers refer to the revised manuscript version with changes tracked. In addition to the changes below, we have also corrected minor typos.*

**Comments by the Editor**

**EC #1:** RC# 2 Major Comment-1 has been adressed in your reply and some changes were made in the text. Please include and discuss the suggested literature by the reviewer in your paper with respect to your site.

*AR #1: We have now included and discussed the works by Atchley et al. (2016), McKenzie and Voss (2013), Helbig et al. (2013), and Sjöberg et al. (2016) in Section 3.3 (page 7, lines 2-6) and Section 5.3 (page 13, lines 7-10).*

**EC #2:** page 24 of your revised paper uses the notations SDD/WDD, as well as root(SDD/WDD). Please make sure that you are using the correct term.

*AR #2: We have removed the instances of sqrt(SDD) and sqrt(WDD) and now use SDD and WDD consistently throughout the manuscript.*

**Report #1 – Comments by Anonymous Referee #2**

Review of revised version of manuscript tc-2016-173

„Soil moisture resdistribution and its effect on inter-annual active layer temperature and thickness variations in a dry loess terrace in Adventdalen, Svalbard" by C. Schuh et al.

General comments:

I have already reviewed the first version of the manuscript. The authors have adequately addressed all of my comments in the revised version, hence I can now recommend the manuscript for publication. I am looking forward to seeing the paper published in The Cryosphere. Please find below a few final minor and technical comments.

Specific comments:

**RC #1:** Throughout the paper, figures and tables: please check spelling of "van Genuchten" and write either small or capital "van".

*AR #1: Corrected to small "van".*

**RC #2:** P 1, L 14: please correct "… is considered…"

*AR #2: Done.*

**RC #3:** P 1, L 17: "… and are subject to…"

*AR #3: Done.*

**RC #4:** P 2, L 3: "… and has implications for…"

*AR #4: Done.*

**RC #5:** P 2, L 9: I suggest to write "… so that the distribution and movement of moisture … is essential…"

*AR #5: Done.*

**RC #6:** P 2, L 24: Here I suggest to write "…, which turned out to be mainly air temperature and solar radiation…"

*AR #6: Done.*

**RC #7:** P 3, L 28: citation Cable et al.: remove citation if not accepted yet

*AR #7: The manuscript by Cable et al. is in final stages of revisions in Geomorphology and we believe it will soon be accepted.*

**RC #8:** P 3, L 38: I think these should be mean monthly GST (?)

*AR #8: Yes; corrected.*

**RC #9:** P 5, L 13: Sentence does not make sense: Richards' equation is the unsaturated version of Darcy's equation, hence "with the unsaturated version of Darcy's equation" should be deleted (if this corresponds with the ATS model equations). Correct spelling: Richards'

*AR #9: The text has been clarified; to avoid ambiguity we refer to the unsaturated version of Darcy's law as a physical/constitutive relationship used to close the system of balance equations.*

**RC #10:** P 6, L 36: better: "… and no-flow boundaries were assigned…"

*AR #10: Done.*

**RC #11:** P 6, L 38: remove reference to Sect 2 and add explanation here for better readability

*AR #11: A phrase has been added to clarify this statement and improve readability (page 7, lines 8-9).*

**RC #12:** P 7, L 36: replace "greatest" by highest"

*AR #12: Done.*

**RC #13:** P 9, L11, 19 and 20: replace "greater" and "greatest" by "stronger" and "strongest"

*AR #13: Done.*

**RC #14:** P 12, L 8: replace "greatest" by "deepest"

*AR #6: Done.*

**RC #15:** P 12, L34-40: This is a very general statement. The soil moisture retention properties may become important as soon as the total water content in the profile changes over time. I suggest to restrict this statement to the particular situation where the total amount of water in the profile remains constant.

*AR #15: Done; we added "In this setting,…" (page 13, line 7).*

**RC #16:** P14, L 4: replace "greater" by "larger" and "stronger"

*AR #16: Done.*

**Report #2 – Comments by Anonymous Referee #1**

I appreciate the work the authors have done to streamline the intro and focus on soil moisture distribution. I think this focusing and the manuscript in general helps with understanding how cryosuction and soil water retention curves form ice lens, redistribute latent heat mass and the affect it has of thaw propagation. However, I should mention that the second to last conclusion, "Active layer thickness, as observed in the field, responded primarily to the cumulative temperature during the preceding winters, as opposed to cumulative summer temperature during thaw" does not seem to be connected to the central aim of studying "how soil moisture retention properties affect moisture and ice redistribution as well as subsurface temperature and active layer thickness variations in the partially saturated active layer under multiple freeze-thaw cycles." Regardless, I feel the manuscript is acceptable for publication.

Also, I should also note that line reference numbers in the response to reviewers was off by a page or two, there is no page 16 in the revised manuscript, including the mark version.

*AR #1: We thank the referee for their careful and constructive review and apologize for any inconvenience caused by the faulty page/line numbering.*

[revised manuscript text omitted]

---

## Author Response (AR3)

**Comments by the Editor**

Editor Decision: Publish subject to technical corrections (18 Jan 2017) by Julia Boike

Comments to the Author:

Dear authors,

thanks again for the revisions.

I have the following corrections:

-Figure 6 shows sqrt(WDD/SDD) in the X-axis labeling, but the figure caption states only WDD/SDD.

-Figure 7 shows sqrt(WDD/SDD) in the y-axis labeling, but the figure caption states only WDD/SDD. Please use the correct units, also corresponding to the text.

Best regard,

Julia

**Author Response**

*We thank the Editor for accepting our ms subject to technical corrections. The previous use of "sqrt(SDD)" and "sqrt(WDD)" in figures 6 and 7 was a typo. We have corrected this to "SDD" and "WDD" respectively in the revised version.*